# Optogenetic control of YAP reveals a dynamic communication code for stem cell fate and proliferation

Kirstin Meyer[1,2], Nicholas C. Lammers[3,4], Lukasz J. Bugaj [5], Hernan G. Garcia [3,6,7,8,9] & Orion D. Weiner [1,2] ✉

YAP is a transcriptional regulator that controls pluripotency, cell fate, and proliferation. How cells ensure the selective activation of YAP effector genes is unknown. This knowledge is essential to rationally control cellular decision-making. Here we leverage optogenetics, live-imaging of transcription, and cell fate analysis to understand and control gene activation and cell behavior. We reveal that cells decode the steady-state concentrations and timing of YAP activation to control proliferation, cell fate, and expression of the pluripotency regulators Oct4 and Nanog. While oscillatory YAP inputs induce Oct4 expression and proliferation optimally at frequencies that mimic native dynamics, cellular differentiation requires persistently low YAP levels. We identify the molecular logic of the Oct4 dynamic decoder, which acts through an adaptive change sensor. Our work reveals how YAP levels and dynamics enable multiplexing of information transmission for the regulation of developmental decision-making and establishes a platform for the rational control of these behaviors.

Morphogenesis relies on accurate cellular decision-making to ensure the correct development of organisms. Despite extensive parts list descriptions of biological systems, our understanding of how cells process and transmit information to ensure robust decision making is limited. Filling this gap will be necessary for the rational control of cell physiology but requires insight into the signaling logic as well as a means to interface with endogenous signaling systems. Here we leverage an optogenetically gated transcription factor to understand and control embryonic stem cell fate and proliferation.

Transcription factors play pivotal roles in the regulation of development. They relay information from the cellular environment to the gene regulatory networks that control cell fate. A striking feature of these transcriptional networks is the use of a relatively small set of

transcription factors to control large arrays of genes during development, with the same transcription factors often regulating different sets of genes in different contexts[1–3]. When a transcription factor is activated, how do cells choose among multiple downstream responses? Addressing this question is not only fundamental for developmental biology but will also be essential for manipulating cell fate for tissue engineering.

YAP (yes-associated protein), the main effector of the Hippo pathway, is a functionally pleiotropic transcriptional regulator. It integrates inputs from cell mechanics[4–6], cell polarity[7], and cell metabolism[8–10] to control the effectors of pluripotency[11,12], germ layer specification (meso-/endo-/ectoderm)[13,14], and proliferation[15,16]. While extensive efforts have identified the signaling modules that act

[1]Cardiovascular Research Institute, University of California, San Francisco, San Francisco, CA, USA. [2]Department of Biochemistry and Biophysics, University of California, San Francisco, San Francisco, CA, USA. [3]Biophysics Graduate Group, University of California at Berkeley, Berkeley, CA, USA. [4]Department of Genome Sciences, University of Washington, Seattle, WA, USA. [5]Department of Bioengineering, University of Pennsylvania, Philadelphia, PA, USA. [6]Department of Physics, University of California at Berkeley, Berkeley, CA, USA. [7]Department of Molecular and Cell Biology, University of California Berkeley, Berkeley, CA, USA. [8]Institute for Quantitative Biosciences-QB3, University of California at Berkeley, Berkeley, CA, USA. [9]Chan Zuckerberg Biohub, San Francisco, CA, USA. ✉e-mail: orion.weiner@ucsf.edu

upstream and downstream of YAP, we lack an operational framework for understanding how YAP controls specific gene regulatory programs and cellular decisions.

In many biological contexts, signaling levels and dynamics are used to specify cellular responses[17,18]. For example, different concentrations of morphogens direct distinct gene regulatory programs, enabling the conversion of continuous morphogen gradients into switch-like boundaries of cell fate[19,20]. Alternatively, the temporal dynamics of signaling inputs (such as the duration, frequency and amplitude) can be used to specify appropriate cellular behavior. For example, ERK and p53 dynamically inform downstream target genes about the identity and magnitude of upstream inputs, respectively. ERK target genes act as persistence detectors to control whether cells proliferate or differentiate[21–23]. p53 pulse frequency conveys the magnitude of DNA damage; this enables cells to decide between DNA repair and cell death[24,25]. Despite fluctuations of YAP levels in developmental contexts[26], it is unclear if cells employ a similar dynamic or concentration-dependent encoding strategy to link patterns of YAP activation to appropriate downstream effectors (Fig. 1a).

Here we engineered mouse embryonic stem cells with an optogenetic YAP tool to investigate how the levels and timing of YAP control downstream target activation and cellular decision-making. Our light-gated YAP signaling strategy provides a mechanism of interfacing with the YAP signaling module, enabling temporal control of nuclear YAP levels and dynamics that cannot be achieved through classical genetic perturbation strategies. By combining our optogenetic system with live imaging of downstream target gene transcription and cell fate readouts, we reveal the input-output logic of YAP signaling. By applying light-gated oscillatory YAP dynamics we demonstrate a dynamic decoding capacity of Oct4, which acts as an adaptive change sensor that optimally engages at specific YAP frequencies mimicking those found in the endogenous system during cellular differentiation. This dynamic decoding mode acts in addition to a steady-state decoding system that reads out overall YAP levels. Analysis of cell fate and proliferation demonstrates that the identified YAP signaling modes also suffice to differentially control cellular differentiation and proliferation. While differentiation requires sustained low YAP concentrations, cell proliferation is most efficiently induced by dynamic YAP inputs. Together, our work reveals how the levels and timing of YAP activation enable multiplexing of information transmission in development and provide a synthetic interface for the control of YAP-dependent stem cell behavior through light. This work further demonstrates the importance of temporal information as communication code in biological systems that can be harnessed to synthetically control cellular behavior.

## Results

### mESCs induce YAP dynamics during differentiation

During differentiation, naive mESCs progressively lose their proliferative capacity and acquire differentiation competence to commit to germ layer fates (Fig. 1b). YAP is a primary determinant of these decisions, including cell fate, pluripotency and proliferation[13,14,27,28]. To assess whether YAP levels or dynamics are modulated during mESCs differentiation, we first analyzed native YAP dynamics in mESCs during pluripotency exit. For this purpose, we generated an endogenous SNAP-YAP reporter mESC line and monitored YAP levels at 1.5d post differentiation. This time point was chosen because it is a generally permissive time window for differentiation cues in mESCs[29]. Live-cell imaging of the meso- and ectodermal lineages (differentiation efficiency > 50%, Supplementary Fig. 1a–c) revealed temporal fluctuations of nuclear YAP levels (Fig. 1c, d, Supplementary Movie 1). While only a small proportion (11%) of naive cells show nuclear YAP fluctuations (Fig. 1e), 36–51% of mESCs differentiating into the ecto- and mesodermal lineages exhibit discrete YAP pulses (Fig. 1e), with an average ~1.5-fold change in amplitude (Supplementary Fig. 1d) that last on average 2.4–2.7 h (Fig. 1f). Quantification of YAP dynamics over the time course of early mesoderm differentiation (Supplementary Fig. 1e) shows that cells slowly induce YAP fluctuations within 1.5d post differentiation start. The dynamics persist throughout at least 3d post pluripotency exit at comparable burst amplitude and duration (Supplementary Fig. 1f, g). This timing coincides with the time window of early differentiation cues, suggesting that YAP dynamics may play a role in the transition from pluripotency exit to early lineage specification. We verified the performance of our YAP peak detection approach (Supplementary Fig. 2a) using both fixed cells stained for endogenous YAP as well as simultaneous live imaging of non-dynamic GFP and endogenous YAP (Supplementary Fig. 2b–h). These controls demonstrate a false positive rate of ~4% (Supplementary Fig. 2f) for our parameter choice. Together, our observation that mESCs exhibit YAP dynamics during pluripotency exit is consistent with the hypothesis that cells employ a temporal decoding strategy for controlling cellular decision making, for example cell fate and proliferation.

### Inducible and light-gated control of nuclear YAP levels in mESCs

To investigate whether the steady-state levels or dynamics of YAP play an instructive role in downstream effector activation, we leveraged both a doxycycline inducible YAP expression system to induce different YAP levels over time scales of days (steady-state) and an optogenetic system to acutely modulate nuclear YAP levels at the time scale of minutes to hours. For the optogenetic approach we adopted a previously reported optogenetic tool termed iLEXYi[30,31] (Fig. 1g) to control nuclear YAP export by light. iLEXYi is based on the AsLOV2 domain[32,33] that exposes a nuclear export sequence upon blue light illumination, resulting in reversible nuclear export. We fused iLEXYi to fluorescently-tagged YAP (iLEXYi-SNAP-YAP termed LEXY-YAP) and also expressed LEXY-YAP under a doxycycline inducible promoter to ensure comparable conditions of our steady-state and dynamic YAP measurements. Our inducible systems were expressed in a YAP KO background to ensure that the entire YAP pool is under expression and/or light control (Supplementary Fig. 3a, b). Doxycycline dose-response analysis demonstrated our ability to achieve a wide range of YAP and LEXY-YAP expression levels that bracket the endogenous YAP levels of WT mESCs (Supplementary Fig. 3c–e). We refer to the inducible system as our method for manipulating steady-state concentrations. In addition, with our optogenetic system, we can achieve rapid and reversible nuclear-cytoplasmic shuttling on the minute timescale (export ~5 min, import ~15 min, Fig. 1h, i, Supplementary Movie 2) with a maximum nuclear YAP depletion of ~60% (Supplementary Fig. 3f). Illumination with different light durations enables pulse width modulation ranging from minutes to hours over extended time periods (10 h; Fig. 1i) that mimic the amplitude and duration of the temporal changes observed in endogenous contexts (see Fig. 1f, Supplementary Fig. 1d). We refer to conditions with continuous light exposure as chronic input with sustained low YAP levels and define conditions with pulsed light exposure as dynamic input with oscillatory YAP dynamics.

We leveraged our inducible YAP and LEXY-YAP tool to investigate steady-state concentration- and time-dependent responses of YAP effectors. In addition to probing YAP levels and dynamics that mimic those observed in the endogenous system, we also investigated a larger parameter space to infer general signaling principles. In the following, we first analyzed how the levels and dynamics of YAP control gene activation. Next, we investigated how downstream effectors decode the levels and timing of YAP activation. Finally, we analyzed how concentration-dependent and dynamic signaling modes control developmental decision-making.

### Steady-state YAP levels differentially control pluripotency factors Oct4 and Nanog

To probe the logic of YAP decoding at the molecular level, we focused on the YAP targets Oct4 and Nanog, which play critical roles in

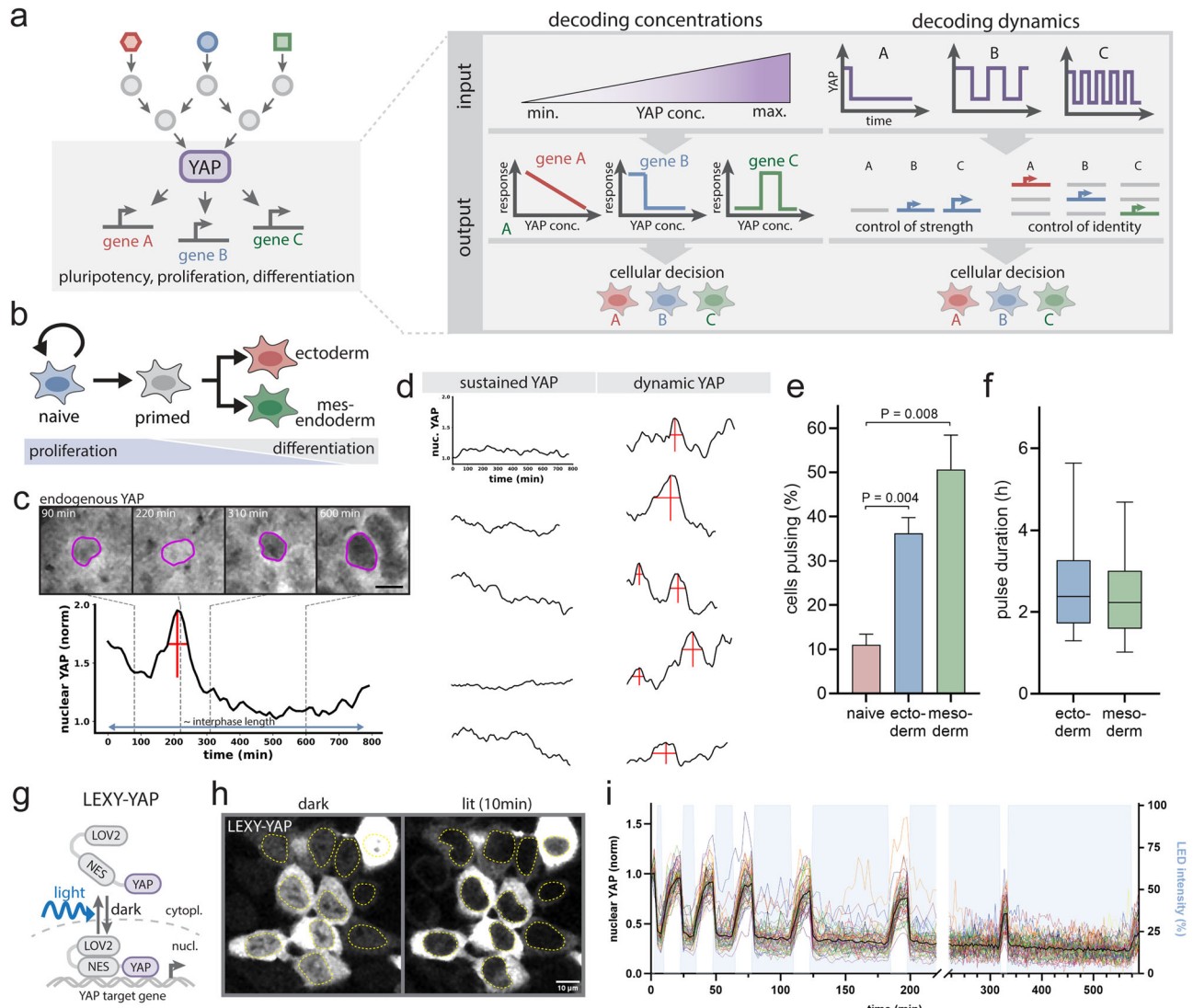

**Fig. 1 | Light-gated control of nuclear YAP levels mimics endogenous YAP dynamics. a** Left: Hourglass-shaped topology of the YAP signaling network. Right: YAP signaling specificity could be achieved through decoding of YAP concentrations and dynamics controlling downstream effector strength and/or identity to direct cellular decisions (e.g., pluripotency, proliferation, differentiation).
**b** Cellular differentiation progresses from a naive stem cell to a differentiation-competent primed stage, and into committed germ layers (ectoderm, mesendoderm). The process is accompanied by progressive loss of proliferative capacity and acquisition of differentiation competence. **c** Pulsatile nuclear dynamics of endogenously tagged SNAP-YAP in interphase mESCs at 36 h post directed mesoderm induction. Shown is the quantification of nuclear YAP levels of a representative cell from $N = 3$ independent experiments. Indicated time points are shown as image on top. The nucleus is outlined (magenta). Scale bar: 10 μm.
**d** Representative single-cell quantification of nuclear YAP levels in the same culture condition as **c**. Traces are classified as sustained or dynamic using an automated peak detection algorithm (see Supplementary Fig. 2). Peak width and height are

indicated by horizontal and vertical red lines. **e, f** Quantification of the percentage of cells exhibiting YAP pulses (**e**) and the average YAP pulse duration (**f**) in mESCs directed along the ectoderm or mesoderm lineages (1.5d post induction). Cells were classified by the peak detection strategy shown in **d** and Supplementary Fig. 2. Shown are mean ± SEM, $N = 3$ independent experiments (**e**), and the Box and Whiskers with median + 5–95 Percentile (**f**). In **f**, data was pooled from $N = 3$ independent experiments. $p$ values from two-sided unpaired Student's t test (**e**).
**g** Optogenetic strategy to control nuclear YAP levels by light using the LEXY-tag: the engineered LOV2 protein domain unfolds a nuclear export sequence (NES) upon blue light illumination, causing reversible nuclear LEXY-YAP export.
**h** Representative images from one experiment of mESCs expressing LEXY-YAP in the dark and following 10 min light illumination. Nuclei are– outlined (yellow). Scale bar: 10 μm. **i** Quantification of nuclear LEXY-YAP levels from microscopy time course demonstrating light-gated induction of consecutive nuclear YAP import/export cycles ranging from the minutes to hours. Blue shading indicate illumination phases, $N = 1$ experiment.

pluripotency maintenance and differentiation[34,35]. Both genes have previously been shown to be regulated by YAP through direct interaction[12,36,37]. We induced steady-state YAP levels and measured the gene dose-response (Fig. 2a) through immunofluorescent (IF) staining of Oct4 and Nanog (Fig. 2b). To cover a wide range of YAP levels, we made use of the expression heterogeneity of the doxycycline inducible system. We analyzed cells expressing SNAP-YAP at 2 days following differentiation and used an undirected differentiation assay (FBS-based media) to establish permissive conditions for instructive signals

from our YAP expression system. Our dose-response experiments identify YAP as a repressor of both Oct4 and Nanog (Fig. 2b,c, see also Supplementary Fig. 4a). Notably, the proteins show different thresholds for inhibition by YAP with IC50s (Fig. 2c, dotted lines) that establish a window for the differential control of Oct4 and Nanog. Within this window, there are levels of YAP that significantly repress Nanog and not Oct4. Above and below this differential control window, YAP acts jointly on Nanog and Oct to permit expression (low YAP) or induce repression (high YAP) of both genes. While we cannot rule

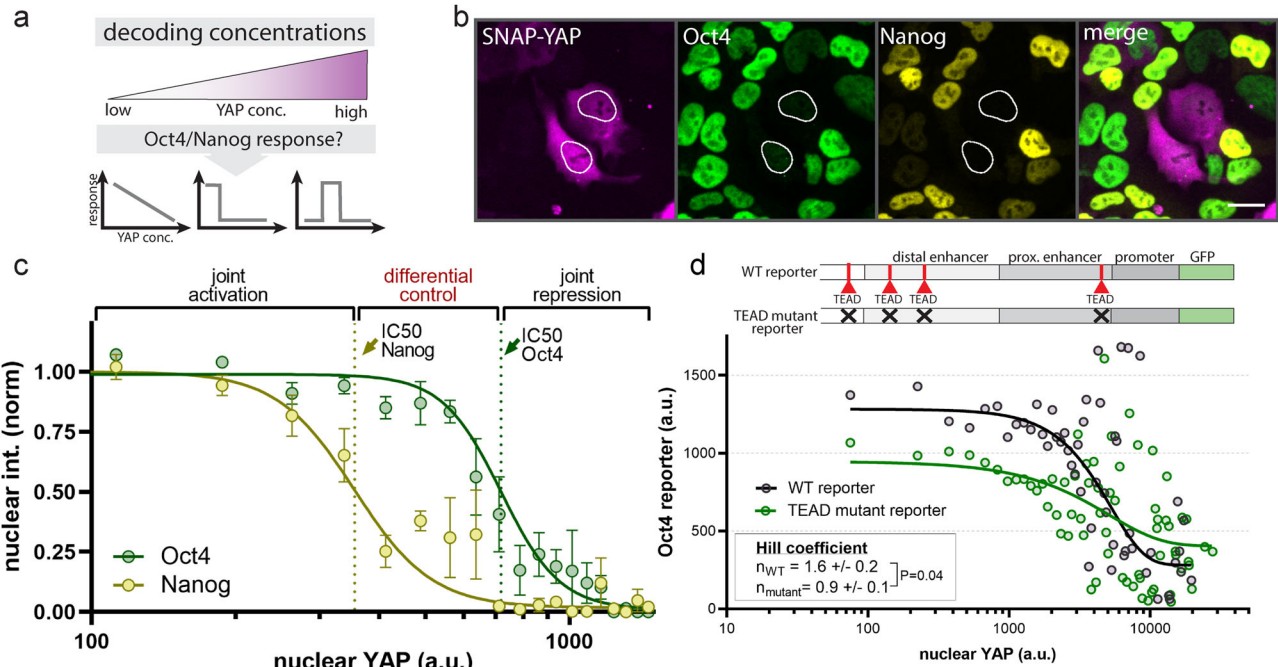

**Fig. 2 | Differential control of Oct4 and Nanog through steady-state YAP concentrations. a** Probing the dose response of Oct4 and Nanog accumulation as a function of steady-state YAP concentrations. **b** SNAP-tag and immunofluorescent staining for SNAP-YAP, Oct4 and Nanog at 48 h post undirected differentiation start shows that YAP is a repressor of Oct4 and Nanog. Nuclei with high SNAP-YAP expression are outlined in white. Scale bar: 20 μm. **c** Sigmoidal curve fit of nuclear Nanog and Oct4 levels as a function of nuclear YAP concentrations reveals differential sensitivity of Oct4 and Nanog to YAP levels through offset repression thresholds (IC50s) that establish three distinct regimes for joint activation of Oct4 and Nanog (low YAP levels), joint repression of Oct4 and Nanog (high YAP levels) and preferential inhibition of Nanog and versus Oct4 (intermediate YAP levels). Dashed lines indicate IC50s. Data was quantified from IF images as representatively shown in **b**. Shown are mean ± SEM, $N = 4$ independent experiments, $R^2 = 0.96$ (Oct4), $R^2 = 0.97$ (Nanog). X-axis is clipped, see Supplementary Fig. 4a for full range. **d** Deletion of the TEAD binding sites in the gene regulatory region of the Oct4 locus reporter affect the sensitivity of the locus to steady-state YAP levels, shifting the Hill coefficient from $1.6 \pm 0.2$ (WT) to $0.9 \pm 0.1$ (TEAD mutant). The Oct4 reporter is ectopically introduced and based on the replacement of the Oct4 open reading frame with GFP (**d**, top). The location of the TEAD binding sites is schematically shown, see Supplementary Fig. 5a for details. Shown are Oct4 reporter levels as a function of nuclear YAP levels and the sigmoidal curve fit from a single experiment. See Supplementary Fig. 5c for replicates. Hill coefficient represents mean ± SEM from $N = 3$ independent experiments. $p$ value from two-sided unpaired Student's $t$ test.

out the occurrence of YAP dynamics under steady-state expression conditions, the evident correlation of nuclear YAP levels with Oct4 and Nanog levels (Fig. 2c, Oct4 $R^2 = 0.96$, Nanog $R^2 = 0.97$) suggest at least a strong concentration-dependency. Naive mESCs are less sensitive to YAP levels with a significant shift of the Oct4 and Nanog IC50s as compared to differentiating cells (Supplementary Fig. 4b), indicating that YAP signaling competence changes during pluripotency exit. Importantly, the observed dose-response behavior is comparable between cells expressing SNAP-YAP and LEXY-YAP (Supplementary Fig. 4c), verifying the functionality of our optogenetic tool.

YAP has functions both in the nucleus and in the cytoplasm. To gain inside into the regulation of Oct4, we dissected the role of nuclear and cytoplasmic YAP for the observed repression of Oct4 protein levels. We made use of the heterogeneity of endogenous nuclear and cytoplasmic YAP levels occurring during spontaneous differentiation (Supplementary Fig. 4d). IF staining of YAP and Oct4 and quantification of nuclear and cytoplasmic YAP shows that nuclear YAP levels are associated with significantly higher Oct4 repression than comparable cytoplasmic YAP levels (Supplementary Fig. 4e, f). This result as well as previous reports on the interaction of YAP with the Oct4 locus suggest that nuclear YAP controls the switch-like repression of Oct4 through regulation of gene expression[12,36,37]. This could be mediated through the presence of TEAD/YAP binding sites in the Oct4 locus (Supplementary Fig. 5a). To test this, we leveraged an Oct4 reporter that harbors the full regulatory region (promoter, proximal and distal enhancer) and in which the Oct4 open reading frame was replaced with GFP (Fig. 2d, schematic on top)[38]. We deleted all four putative TEAD binding sites in the regulatory region and repeated our steady-state

quantification of Oct4 repression using our doxycycline inducible system. While YAP repressed both the WT and TEAD mutant reporter, the absence of the TEAD sites rendered the reporter more sensitive to low YAP levels and almost entirely diminished the switch-like repressive behavior (Fig. 2d, WT Hill coefficient= $1.6 \pm 0.2$; TEAD mutant reporter Hill coefficient = $0.9 \pm 0.1$; see also Supplementary Fig. 5b, c). Together, our results demonstrate the differential control of the pluripotency factors Oct4 and Nanog through steady-state YAP levels, analogous to the role of morphogens in determining cell fate in a concentration-dependent manner. Our work reveals that the switch-like behavior of Oct4 repression is established through the TEAD/YAP binding sites in the Oct4 regulatory region, suggesting a negative cooperative effect of YAP/TEAD binding. We next sought to investigate if the dynamics of YAP control gene activation using our optogenetic approach.

**The pluripotency factor Oct4 decodes YAP dynamics**
To test how the timing of YAP activation modulates Oct4 and Nanog regulation, we used our LEXY-YAP tool to dynamically control the concentration of YAP in the nucleus. While we apply oscillatory light patterns here, we do not imply that endogenous YAP dynamics are oscillatory but rather occur as sporadic pulses (see Fig. 1c, d). The light-gated oscillatory YAP dynamics include conditions with constant export duration (4 h) but varying recovery (import) periods for a total of 12 h during the onset of differentiation (24–36h post pluripotency exit; Fig. 3a). We investigated YAP pulse durations ranging from 15 min to 4 h that bracket the range of temporal pulse features observed in the endogenous system (see Fig. 1f). IF staining

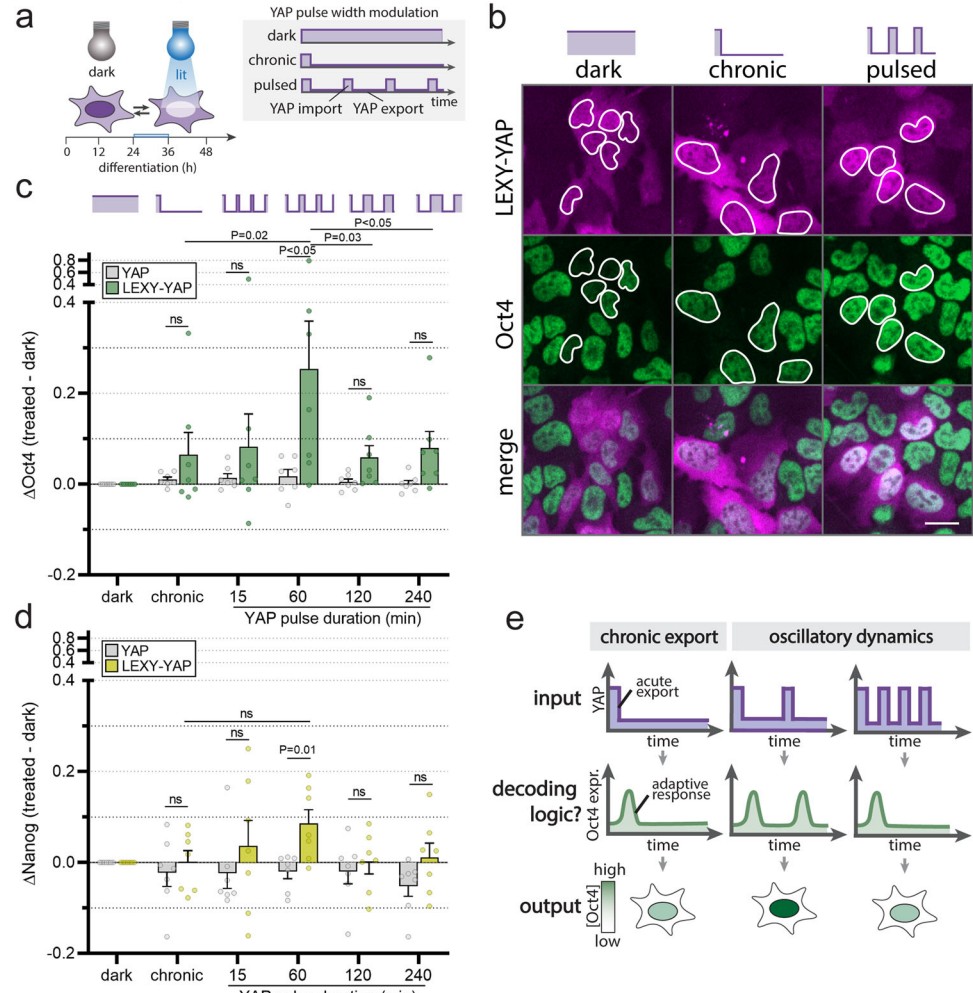

**Fig. 3 | Oct4 and Nanog respond to YAP dynamics. a** To probe whether YAP targets are sensitive to dynamic YAP inputs, ES cells expressing LEXY-YAP were exposed to light patterns with different pulse widths during pluripotency exit. **b** IF staining for Oct4 in LEXY-YAP mESCs upon illumination with chronic or pulsed light inputs demonstrates induction of Oct4 upon illumination as compared to the dark control, with higher Oct4 levels in the oscillatory than chronic YAP export condition. Scale bar: 20 μm. **c, d** Quantification of nuclear Oct4 (**c**) and Nanog (**d**) levels from IF staining as shown in **b**. Cells were subjected to light pulses inducing constant YAP export durations of 4 h and varying import duration ranging from 15 min to 4 h as indicated. Only cells with moderately high LEXY-YAP/YAP expression levels (>IC50 of dark control) were quantified (see Supplementary Fig. 6b for comparison to cells with lower YAP expression levels). Pulse frequencies with specific YAP export/import cycles (4 h export, 60 min import) significantly upregulate Oct4 and Nanog as compared to the non-light responsive YAP control or compared to

chronic YAP export. Quantifications shown only include cells with YAP levels >= IC50 in the dark condition. Note that the chronic light condition is sufficient to induce Oct4 in cells with lower YAP levels (see Supplementary Fig. 6a, b). Note that the induction of Nanog by oscillatory inputs is minor. Shown are mean ± SEM, $N = 7$ independent experiments, $p$ values comparing LEXY-YAP vs YAP from two-sided unpaired Student's t test; $p$ values comparing light conditions within the LEXY-YAP group from two-sided paired non-parametric t test. **e** Possible decoding logic of dynamic YAP inputs through adaptation. For an adaptive system, continuous YAP export would transiently activate YAP effectors (left panel). In contrast, pulsed YAP inputs would induce sequential adaptive YAP responses; this would result in higher total Oct4 induction (center panel) than is seen for chronic YAP export (left panel). The adaptive system gives a maximum output at a specific pulse frequency input (center) compared to faster (right panel) or slower (not shown) frequencies.

and single-cell quantification of Oct4 protein levels revealed that Oct4 generally shows a minor response to chronic YAP export (Supplementary Fig. 6a, top left graph) but a potent response to oscillatory YAP dynamics (Supplementary Fig. 6a, top right graph). While chronic light only induces Oct4 at lower YAP levels (Supplementary Fig. 6b for cells expressing YAP levels >=IC5 of dark control), oscillatory light suffices for Oct4 induction even at high levels of YAP (Fig. 3c for cells expressing YAP levels >=IC50 of dark control) at a magnitude that is significantly different from the chronic response. Among the oscillatory light inputs, some patterns of YAP dynamics induce significantly higher Oct4 levels than others, demonstrating a dynamic filtering capacity. The system reaches a maximum response upon illumination with a 4 h export/1 h recovery duty cycle (Fig. 3c).

The observed filtering capacity of Oct4 to specific oscillatory YAP inputs suggests that Oct4 decodes YAP dynamics. However, these experiments do not distinguish decoding modes that read out dynamics features versus those that read out a specific level of integrated YAP over time. To probe this question, we monitored the Oct4 response upon illumination with dynamic light patterns that differ in pulse duration but use the same total light input (pulse modulation, Supplementary Fig. 7b, top panel). In addition, we apply a chronic light dose titration to monitor if the integrated YAP export in absence of pulsatile dynamics can explain our results (Supplementary Fig. 7b, bottom panel). For the chronic light titration, we established the YAP dose response to different light intensities by YAP IF (Supplementary Fig. 7a). Our results demonstrate that a time-integrated decoding mode of Oct4 cannot explain our results. We find that Oct4 responses

differ significantly when exposed to different pulse durations that share the same total integrated light inputs (Supplementary Fig. 7c, pulse modulation). In line with this, chronic light doses comparable to the total light input of the dynamics conditions are insufficient to induce Oct4 (Supplementary Fig. 7c, light dose modulation). Importantly, the Oct4 response (Supplementary Fig. 7c) was measured across the YAP range where the full chronic light input shows a response (>IC5 of the dark control, see Supplementary Fig. 6a). Altogether, our results demonstrate that the Oct4 signaling module has two different decoding capacities for steady-state and dynamical inputs. The dynamic decoder operates optimally at 1 h YAP pulse duration and 4 h reset time, a range that brackets endogenous YAP dynamics.

To relate the magnitude of our Oct4 phenotypes to physiological Oct4 levels, we used Oct4 IF staining to compare optogenetic and steady-state vs native Oct4 modulation (Supplementary Fig. 6c). Following pluripotency exit, mESCs decrease endogenous Oct4 levels by ~73% over a time course of 4 days post spontaneous differentiation. Our steady-state YAP expression system (Fig. 2c) acts on Oct4 levels across most of the range of the endogenous system (Supplementary Fig. 6c). The magnitude of the observed Oct4 induction upon dynamic YAP inputs (ΔOct4 = 0.25, Fig. 3c) is comparable to the amount of Oct4 protein lost within ~1d during spontaneous differentiation (Supplementary Fig. 6c). These results demonstrate the physiological relevance of the observed Oct4 regulatory modes.

We next compared the Oct4 response to Nanog. For the same LEXY-YAP expression levels, Nanog shows a response to dynamic inputs that follows the same pattern (Fig. 3d) but at substantially lower magnitude than Oct4. Together and in conjunction with our steady-state experiments, our data show that both the levels and timing of YAP activation differentially engage the YAP downstream targets Oct4 and Nanog through different regulatory modes. While steady-state levels differentially control Oct4 and Nanog within a fixed concentration regime, dynamic YAP inputs jointly induce Oct4 and Nanog, although with different magnitudes. Given the importance of transcription factor ratios, for example during cellular reprogramming, this difference in magnitude could establish differential control regimes similar to the threshold-dependent decoupling of Oct4 and Nanog expression under steady-state YAP concentrations. Taken together, these identified regulatory modes could provide a means for the complex regulatory requirements of Oct4 and Nanog: pluripotency maintenance requires overlapping high Oct4 and Nanog expression (achievable at low YAP), while mesodermal lineage induction requires mutually exclusive control (low Nanog and high Oct4, achievable at intermediate YAP or dynamic YAP inputs) or overlapping low expression (low Nanog and low Oct4, achievable at high YAP)[35,39].

Our data show optimal activation of YAP targets at a particular frequency of YAP activation. What is the basis of this temporal decoding? One potential mode of preferentially responding to a given signaling dynamics is the adaptive circuit found in other systems such as chemotaxis[40], osmoregulation[41], sensory systems[42,43], and other transcription factors[44,45]. In response to an acute chronic input, adaptive signaling circuits transiently respond to the change in signal input but then reset to the initial baseline under sustained activation of the input (Fig. 3e, left panel). Adaptive systems require resetting between rounds of activation and only produce a single pulse for oscillatory inputs that are too closely spaced in time (Fig. 3e right panel). As a result, adaptive systems generate maximal responses at a specific input frequency that matches the resetting time (Fig. 3e, center panel). We next used Oct4 to investigate whether YAP targets make use of an adaptive strategy to decode the temporal dynamics of YAP.

### Oct4 acts as an adaptive change sensor of YAP

To investigate how Oct4 expression responds to acute changes in YAP, we leveraged the MS2 system to visualize real-time transcription in individual living cells[46]. The MS2 system is based on the integration of a repetitive MS2 DNA array into the endogenous gene locus (Fig. 4a). Transcription of the array generates hairpin structures at the RNA level that are detectable through recruitment of a Halo-tagged coat protein. Local spots of labeled RNA are visible at the transcription site and provide a measure of transcriptional activity of the gene locus. We integrated the MS2 array into the endogenous Oct4 locus in WT and YAP KO background mESCs and observed sporadic bursts of Oct4 RNA production (Fig. 4b) that are characteristic features of transcriptional activation. The MS2 spots colocalized with the Oct4 locus by DNA FISH (Supplementary Fig. 8a). To verify that YAP is controlling Oct4 at the gene regulatory level, as expected for a transcriptional regulator, we compared the Oct4-MS2 reporter in YAP KO to WT cells at 2d following pluripotency exit. While WT cells show moderate transcriptional activity (on average 18 ± 4.2% (mean ± SEM) of the cells are transcriptionally active), depletion of YAP strongly potentiates Oct4 expression, with an average of 62 ± 4.6% (mean ± SEM) cells bursting (Supplementary Fig. 8b, c; Supplementary Movie 3). This confirms the role of YAP as a transcriptional repressor of Oct4.

To analyze how Oct4 transcriptional activity responds to steady-state YAP levels and dynamics, we used the Oct4-MS2 reporter to monitor transcriptional activity before and after acute light-gated nuclear export. Mapping the dark (pre-activation) phase single-cell Oct4-MS2 activity to steady-state nuclear LEXY-YAP levels, we found a similar concentration-dependent repressive effect (Fig. 4c), as observed at the protein level (see Fig. 2c). These data demonstrate that the steady-state control of Oct4 is implemented at the level of transcription. Co-staining of LEXY-YAP cells and WT mESCs for YAP reveals that the dose-responsive regime of the LEXY-YAP tool falls into the endogenous YAP concentration range of WT cells (Fig. 4c, gray shading), indicating that our LEXY-YAP tool operates in a physiological range.

Next, we analyzed the Oct4-MS2 behavior after light-gated YAP export from the nucleus. Considering only LEXY-YAP cells with expression levels in the endogenous range (see Fig. 4c), we observe an adaptive response of Oct4 transcriptional activity, with a transient peak of activation followed by return to baseline levels (Fig. 4d, top panel; Supplementary Movie 4) in the presence of sustained YAP export (Fig. 4d, bottom panel). The response is only detectable in cells that are actively bursting when initiating the acute YAP export (Supplementary Fig. 8e), which would be consistent with a role for YAP in enhancing but not initiating locus activity. The Oct4 MS2 response exhibits an approx. 45 min delay relative to the time of initial illumination and lasts for about 1.5 h before resetting to the baseline. This adaptive response is different from a concentration-dependent mode of regulation, in which we would expect Oct4 to be persistently activated following sustained YAP export from the nucleus. The time scale of the Oct4 response (45 min delay to light onset and 1.5 h duration of Oct4 expression) is consistent with our observations at the protein level; these respond to light pulses of 4 h pulse width to efficiently increase Oct4 protein levels. Our results reveal that Oct4 employs an adaption-based decoding mechanism of YAP dynamics with a characteristic response time on the hour time scale that is reminiscent of the dynamics observed for endogenous YAP during directed differentiation.

### YAP regulates Oct4 transcription through modulation of burst frequency

Our results reveal two different YAP decoding modes by Oct4 that are implemented at the gene regulatory level: steady-state YAP levels control Oct4 in a dose-dependent manner, while acute YAP changes induce an adaptive transient response. How nuclear YAP concentrations or dynamics are sensed ("decoded") by the Oct4 gene is unclear. Here, we set out to test how YAP interfaces with the underlying gene regulatory network.

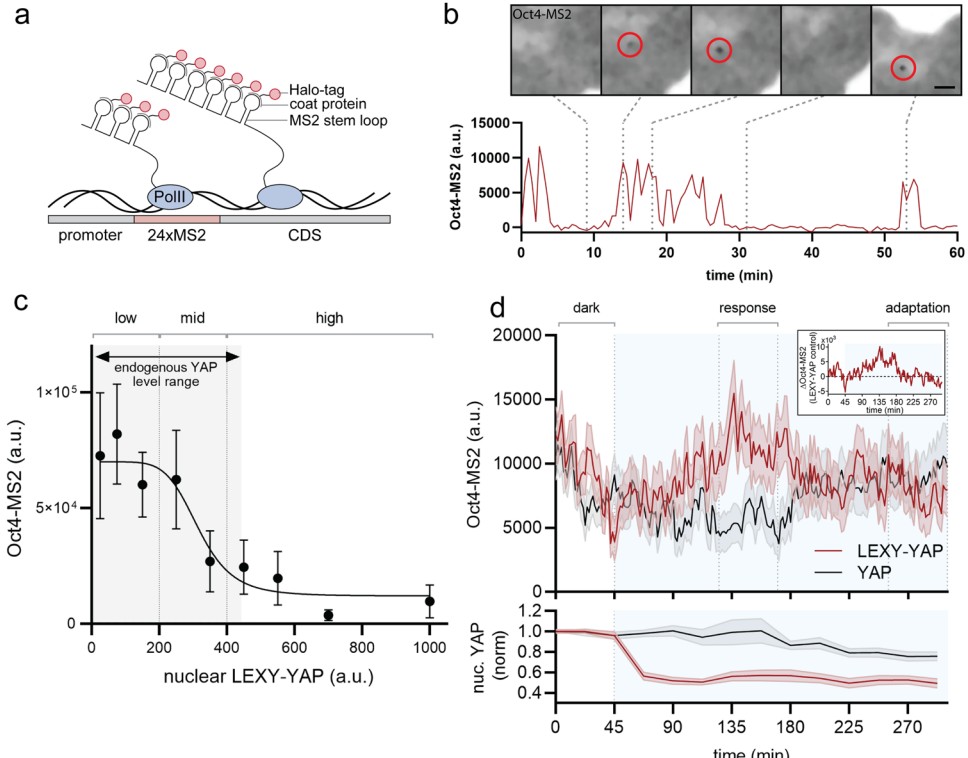

**Fig. 4 | Oct4 acts as adaptive change sensor of YAP levels. a** MS2 system for visualization of transcription in single living cells. Transcription of the 24xMS2 DNA array generates RNA stem loops that are detected through their recruitment of Halo-tagged coat proteins and visible as fluorescent spots at the transcription site (**b**). CDS: Coding sequence. **b** Oct4-MS2 live-imaging reporter in WT mESCs shows transcriptional bursting. Shown are example microscopy data for indicated time points (dashed line) of the time course. Scale bar: 2.5 μm. **c** Oct4-MS2 signal as a function of steady-state nuclear LEXY-YAP levels reveals that YAP acts as a repressor of Oct4 transcription. Min-max range of endogenous YAP levels measured in

WT cells is indicated by gray shading. Shown are mean ± SEM, $N = 15$ independent experiments. **d** Top: Oct4-MS2 transcriptional activity upon light-gated nuclear YAP export in LEXY-YAP mESCs (red curve) vs. non-light responsive YAP control cells (gray curve). Difference between mean of LEXY-YAP and YAP curve is shown as inset on the top right, indicative of transient adaptive YAP target activation in response to sustained export of YAP from the nucleus. Bottom: Nuclear YAP levels in LEXY-YAP (red) and YAP (gray) mESCs simultaneously imaged for Oct4-MS2 shown in the top panel. Illumination phase is indicated by blue shading. Shown are mean ± SEM, $N = 12$ independent experiments.

As shown in our single-cell Oct4-MS2 trace (see Fig. 4b), transcription occurs as sporadic bursts of nascent RNA synthesis. These bursts are shaped by the transcription cycle in which gene promoters switch from an OFF to an ON state to initiate polymerases[47,48]. To shape gene expression, transcriptional regulators such as YAP can interface with any steps in this cycle to modulate the total RNA output by control of burst initiation, duration, and amplitude. For example, YAP could act through chromatin modification or recruitment of polymerases to alter transcription. The different YAP signaling modes observed (steady state vs dynamic) could either be established through differences in how YAP interfaces with the transcriptional cycle or through different YAP regulatory modes that act independent of the transcription regulatory logic. Biochemical snapshots of YAP interaction partners and chromatin state are insufficient to reveal the regulatory entry points shaping the dynamic process of YAP-mediated transcription. Here we investigate the decoding logic of YAP concentrations and dynamics by applying a previously-reported theoretical approach to infer promoter states from our experimental transcription traces.

RNA polymerases take minutes to read through a gene locus. The MS2 signal recorded by our live imaging approach therefore only reflects an integrated readout of transcriptional activity over the elongation time. To deconvolve the instantaneous promoter states from the MS2 signal, we use a previously-reported compound-state Hidden Markov Model[49] that describes transcriptional bursting as a stochastic process in which the promoter switches between an ON and OFF state with $k_{on}$ and $k_{off}$ rates, and initiates polymerases at rate $r$ when in the ON state (Fig. 5a top panel). The rate constants define

transcriptional features and enable us to infer burst frequency ($k_{on}$), duration ($k_{off}$) and amplitude ($r$; Fig. 5a bottom panel) to compare principles of gene regulation under different YAP inputs (concentration vs dynamics).

First, we investigated the molecular logic of how steady-state YAP levels mediate gene repression of Oct4. To this end, we compared Oct4 burst features in YAP-depleted mESCs to those of WT cells. Inference analysis reveals that YAP depletion increases Oct4 burst frequency and amplitude by ~13-fold and ~4.4-fold as compared to WT cells, respectively (Fig. 5b). While the molecular mechanisms of burst frequency and amplitude modulation are unclear, the fact that both parameters are altered upon YAP perturbation suggests that YAP may control Oct4 expression through more than one regulatory entry point.

Next, we compared the transcriptional burst modulation between our Oct4 steady-state dose response (Fig. 4c) compared to the adaptive response following acute YAP export (see Fig. 4d). To this end, we classify cells into low, middle and high YAP expressors for the dose-response experiments (see Fig. 4c) and into dark or early and late lit phase for the temporal analysis of the adaptive response (see Fig. 4d). Interestingly, and consistent with the YAP knockout phenotype, both YAP decoding modes (steady-state and dynamic YAP) are established through the same transcription regulatory logic: the regulation of Oct4 burst frequency (Fig. 5c, d). While burst frequency gradually decreases with higher YAP levels (Fig. 5c), the adaptive Oct4 response is based on a transient ~3.5-fold increase in burst initiation rate as compared to the control (Fig. 5d). Consistent with the adaptive nature of the MS2 signal, burst frequency almost fully resets to the dark-state baseline after 4 h.

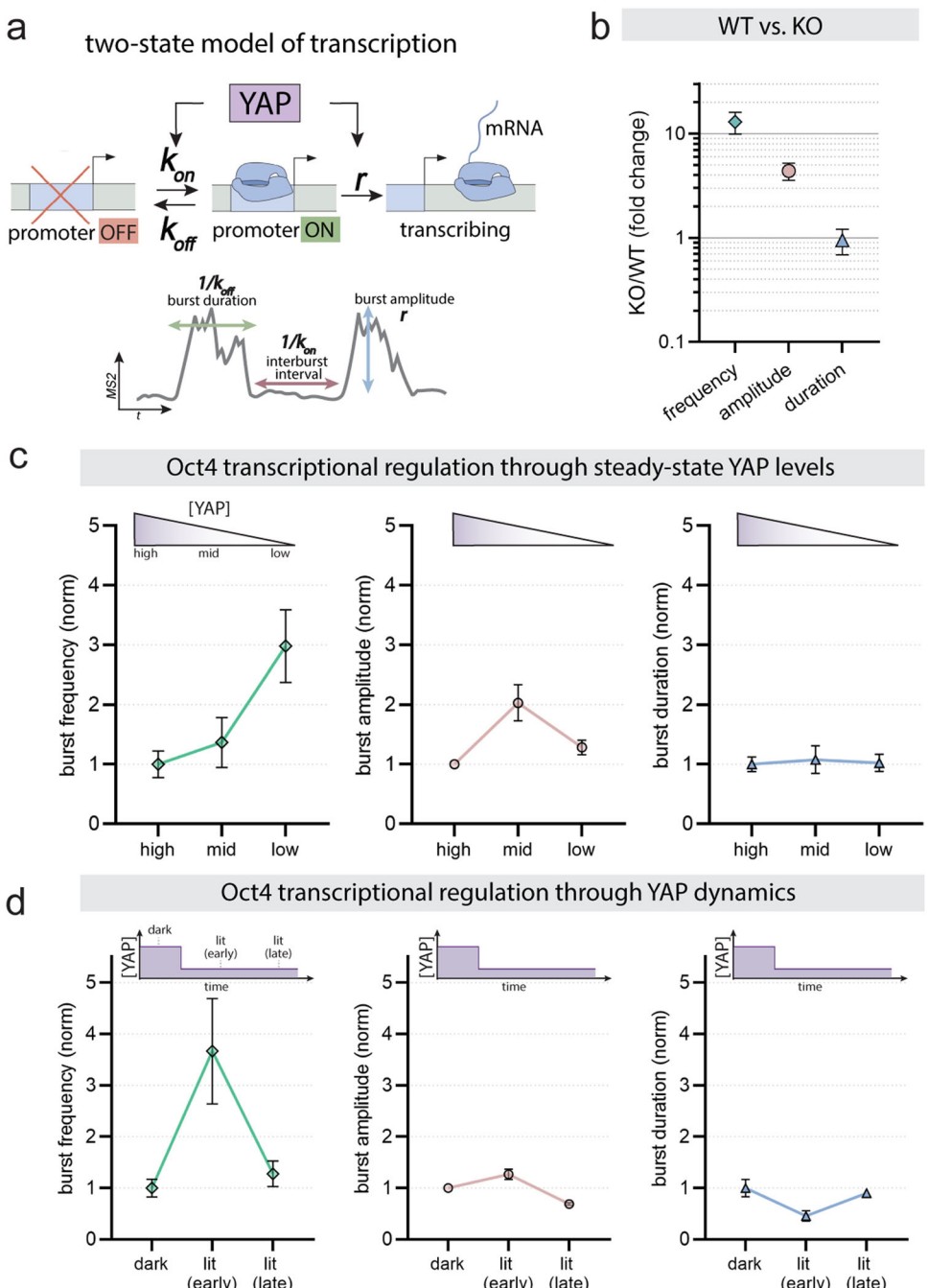

**Fig. 5 | YAP regulates Oct4 transcription through modulation of burst frequency. a** Top: Two-state model of transcriptional regulation. We used this simple model to investigate where YAP interfaces with the transcription cycle to regulate downstream effectors with rate constants $k_{on}$ for activation of the promoter, $k_{off}$ for inactivation of the promoter, and $r$ for transcriptional initiation. Bottom: The rate constants determine transcription burst duration, inter-burst interval, and burst amplitude. **b** Compound-state Hidden Markov Model-based inference of transcription burst parameters (frequency, duration, amplitude) for Oct4 in YAP KO cells, normalized to WT. p values comparing KO vs WT: frequency (Oct4: $P < 0.002$), amplitude (Oct4: $P < 0.002$) and duration (Oct4: $P = 0.47$). **c, d** Transcription burst parameters inference results for Oct4 in response to titration of steady-state YAP levels (**c**) or upon acute nuclear YAP export (**d**). High, mid and low YAP levels relate to indicated YAP concentration regimes in Fig. 4c. Dark, lit (early) and lit (late) relate

to phases indicated as dark, response and adaptation in Fig. 4d, respectively. For **c**, p values comparing high to mid or high to low YAP levels: frequency (mid: $P = 0.21$; low: $P < 0.002$), amplitude (mid: $P < 0.003$; low: $P = 0.007$), duration (mid: $P = 0.41$; low: $P = 0.44$). For **d**, p values comparing lit (early) vs dark or lit (late) vs dark: frequency (lit early: $P = 0.003$; lit late: $P = 0.29$), amplitude (lit early: $P = 0.1$; lit late: $P = 9.3 \times 10^{-5}$), duration (lit early: $P = 0.09$; lit late: $P = 0.39$). Error bars in **b**–**d** reflect the standard error of the mean, as estimated from no fewer than 16 cpHMM inference replicates conducted on bootstrap samples of the experimental data. Inference analysis was performed on pooled data from $N = 4$ (**b**), $N = 15$ (**c**), and $N = 12$ (**d**) independent experiments. All $p$-values were calculated using 1-sided tests based on the comparison of multiple bootstrap replicates. See Methods for a detailed description of how $p$-values were estimated.

As for the knockout phenotype, both the steady-state and the adaptive response are accompanied by a small increase in burst amplitude. Together, the results demonstrate that the adaptive change sensor and dose response module affect the same transcription regulatory mechanism (modulation of burst frequency and amplitude). This suggests that observed differences in the interpretation of steady-state and dynamic YAP inputs are not established through differences in the transcription regulatory logic of the Oct4 locus.

### YAP levels and dynamics differentially control mESC differentiation and proliferation

We have shown that individual YAP target genes differentially decode YAP levels and dynamics. Next, we investigated how YAP levels and dynamics orchestrate integrated cellular responses including lineage specification and proliferation using our established inducible and light-gated YAP control systems during mESC differentiation.

To test the role of YAP concentrations for germ layer specification and proliferation, we induced a range of YAP expression levels using our doxycycline system (see Supplementary Fig. 3d,e) and monitored differentiation induction, lineage specification, and proliferation during spontaneous differentiation. To ensure permissive differentiation conditions for all three germ layer fates, we chose two different spontaneous differentiation media. We used the same FBS-based differentiation media as in our previous experiments (Figs. 2–4). This condition favors mesendoderm differentiation, yielding on average 26% mesoderm (Tbra positive) and 14% endoderm (FoxA2 positive) specification in WT mESCs. In addition, we used N2B27 media, which is permissive for the ectoderm fate, yielding on average 28% ectoderm (Sox1 positive) specification for WT mESCs at 5d post differentiation. First, to read out differentiation induction, we used immunostaining to visualize the early differentiation marker Otx2 at 2 days post differentiation initiation. Otx2 marks cells that have exited pluripotency and are in the transition state to lineage specification[50]. Comparison of nuclear YAP and Otx2 intensities revealed a negative correlation (Supplementary Fig. 9a; see the dark control condition in Supplementary Fig. 10a for a quantitative dose response), demonstrating that low to moderate YAP levels are required to permit differentiation, while high YAP levels inhibit differentiation. To distinguish if YAP generally induces differentiation or acts as specific lineage determinant, we assayed the germ layer markers Tbra (mesoderm), FoxA2 (endoderm) and Sox1 (ectoderm) at 5 days post spontaneous differentiation start for a range of YAP levels (Supplementary Fig. 9b, c). Quantification of lineage markers revealed that YAP specifically acts as repressor of the mesendoderm fate but not the ectodermal fate (Fig. 6a). While low or zero YAP levels efficiently induce mesendoderm, increasing steady-state YAP levels progressively impair fate specification by up to ~90% compared to the WT control. This dose response is absent for the ectodermal fate, which differentiates with similar efficiency as the WT control irrespective of the YAP levels (Fig. 6a). The mesendoderm dose response is consistent with that of Oct4 (see Fig. 2c), which is known to be a primary regulator of the mesendodermal fate. These data demonstrate that YAP acts at the level of early differentiation as well as the level of mesendoderm specification.

We next assayed YAP's control of cell proliferation. For our FBS-based spontaneous differentiation media, we do not detect differences in cell numbers for the different steady-state YAP concentrations (Fig. 6b). To test if YAP's reported pro-proliferative function is compensated by the growth factor-containing serum, we compared this condition to the serum-free differentiation media (N2B27; Supplementary Fig. 9d, compare dashed vs. solid line). In absence of serum, our results demonstrate that YAP levels positively correlate with cell numbers (Supplementary Fig. 9d, dashed line), opposing its effect on cellular differentiation. Importantly, the mesendoderm dose-response (see Fig. 6a) was detected in serum-containing media, demonstrating that the observed differentiation phenotypes are not a consequence of

cell density. Together, we find a YAP concentration-dependent signaling logic for cellular-decision making that establishes concentration regimes for the differential control of both differentiation (mesendoderm) as well as proliferation. While low or high YAP levels only promote differentiation or proliferation, respectively, intermediate YAP levels provide a concentration window for the joint control of both cellular decisions.

Finally, we tested the role of YAP dynamics for mESC differentiation and proliferation (using the FBS-based spontaneous differentiation media). To this end, we used our LEXY-YAP tool to induce light-directed oscillatory YAP dynamics. To verify the functionality of the LEXY-YAP tool for cellular differentiation, we compared the steady-state dose response curve of YAP (see Fig. 6a) to LEXY-YAP (Supplementary Fig. 9e). Both constructs elicited comparable effects on germ layer specification (mesendoderm repression; ectoderm unaffected). We induced YAP dynamics with constant export duration (4 h) but varying recovery (import) periods for a total of 24 h during the onset of differentiation (24h-48h post pluripotency exit) and screened YAP pulse durations ranging from 15 min to 4 h (Fig. 6c, d) that bracket the range of temporal pulse features seen in the endogenous system (see Fig. 1f). Consistent with the dose response curve, IF staining and single-cell quantification of the early differentiation marker Otx2 revealed that differentiation requires persistently low YAP levels (Fig. 6c, Supplementary Fig. 10a,b). While continuous YAP export significantly induces Otx2 intensity compared to the dark control (Fig. 6c line graph, Supplementary Fig. 10a,b), oscillatory YAP activation diminishes the effect for all pulse durations tested (Fig. 6c bar graph inset). Interestingly, the proliferative response reveals a complementary decoding logic to that of cellular differentiation. Although both differentiation and proliferation respond to chronic YAP export, proliferation was most efficiently induced upon oscillatory YAP inputs (1.6-fold) with short (15 ± 60 min) YAP pulse durations (Fig. 6d). This demonstrates a dynamic YAP signaling code that communicates with the proliferation but not with the differentiation module. Interestingly, long YAP pulse durations (4 h) are insufficient to significantly increase cell numbers, mimicking the dynamic filtering capacity of Oct4. Together, our results demonstrate that the modulation of YAP levels and dynamics are sufficient to differentially instruct cellular differentiation (requires chronic low YAP levels) and proliferation (requires oscillatory YAP dynamics), providing an operational framework for understanding how YAP controls specific cellular decisions.

## Discussion

Early embryonic development, recapitulated in mESCs, requires a tight balance between pluripotency maintenance and the induction of differentiation competence; this balance enables cells to interpret lineage specifying signals (Fig. 7a). YAP is a master regulator of these cellular decisions, including pluripotency, proliferation, and differentiation[11–16]. How cells achieve signaling specificity to map the right input to the right gene regulatory programs through the single node of YAP has remained an open question. Here we investigate how cells decode YAP levels and timing by using an optogenetic approach to directly manipulate YAP. Our work reveals both concentration-dependent and dynamic signaling modes of YAP in the control of gene activation (Oct4, Nanog), cell proliferation and differentiation (Fig. 7b, c).

We identify dose dependent interpretations of steady-state YAP levels that direct pluripotency factors Oct4 and Nanog, cellular differentiation, and proliferation (Fig. 7b). Because Oct4 and Nanog have different thresholds of YAP for repression, cells can titrate YAP doses for either joint or mutually exclusive regulation of Oct4 and Nanog. This provides a means for their complex regulatory requirements for pluripotency maintenance (high Oct4 and Nanog) and differentiation (low Nanog, high or low Oct4; Fig. 7b). In addition, YAP levels shape two opposing but overlapping gradients of cellular differentiation (low YAP) and proliferation (high YAP) enabling their balanced control. This

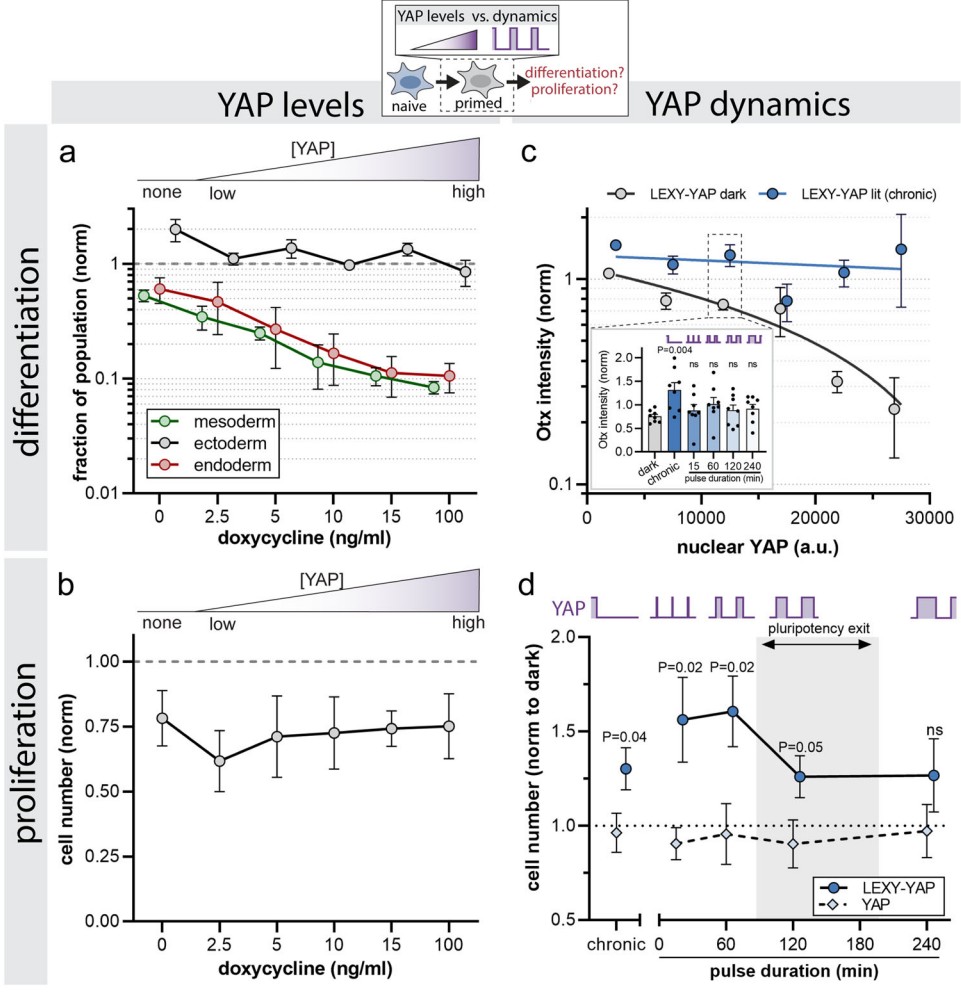

**Fig. 6 | YAP levels and dynamics differentially control cellular differentiation and proliferation. a** Here we investigated how the steady-state levels of YAP control specification of the mesoderm, endoderm, and ectoderm lineages. mESCs were grown under spontaneous differentiation conditions (FBS-based or N2B27, see Methods), and cell types were determined by lineage markers from IF images as shown in Supplementary Fig. 9b, c. YAP expression was induced by titration of doxycycline at indicated concentrations (see Supplementary Fig. 3c, e). Data is normalized to doxycycline treated WT cells. Mesendoderm but not ectoderm respond to steady-state increases of YAP. Shown are mean ± SEM, $N = 4$ independent experiments. Two-sided Student's t test comparing 0 ng/ml dox vs. 100 ng/ml dox, P = 0.02 (endoderm), $P = 0.0004$ (mesoderm), $P = $ n.s. (ectoderm). **b** Quantification of cell number as a function of steady-state YAP levels. Spontaneous differentiation in FBS-based media, YAP induction conditions are the same as described for (**a**). mESC proliferation shows no correlation to YAP levels under these conditions. Shown are mean ± SEM, $N = 4$ independent experiments. Two-

sided Student's t test comparing 0 ng/ml dox vs. 100 ng/ml dox: $P = $ n.s.
**c**, **d** Quantification of differentiation induction (**c**) and cell number (**d**) of LEXY-YAP cells upon chronic and pulsed light illumination in FBS-based differentiation media. Light pulses induce constant YAP export durations of 4 h and varying import duration ranging from 15 min to 4 h as indicated. Persistent YAP export (chronic light) induces Otx2 expression (line graph, blue vs gray line) more efficiently than pulsed light conditions (bar graph inset). In contrast, proliferation is most efficiently induced upon dynamic illumination with short (15-60 min) YAP import cycles (**d**). Gray shadings in **d** indicate pulse durations (lower to upper quartile) of YAP dynamics observed during pluripotency exit (see Fig. 1f). Shown are mean ± SEM, $N = 8$ independent experiments. $p$ values shown in **c**, **d** are from two-sided unpaired Student's t test comparing lit vs. dark condition (**c**) and LEXY-YAP vs. YAP for each illumination condition (**d**). t-tests are not corrected for multiple comparison.

could enable the transition from highly proliferative stem cells to less proliferative and differentiating somatic cells. We further demonstrate that cells are also sensitive to the timing of YAP activation. Oscillatory dynamic YAP inputs more efficiently induce Oct4 and proliferation than do sustained inputs (Fig. 7c). By applying a range of YAP activation frequencies (min-hour time scale) that bracket endogenous dynamics in mESCs, we find that natural dynamics fall into the optimum frequency-decoding range, suggesting they represent physiological communication codes. Importantly, differentiation induction is insensitive to these dynamic inputs, providing a YAP communication code that specifically induces Oct4 and proliferation. Together, our results identify cellular signaling strategies to achieve the complex regulatory requirements of Oct4 and Nanog and to differentially control proliferation and differentiation during development.

While YAP has been established as a central determinant of development, its exact regulatory function has remained elusive due to conflicting reports on its requirement for pluripotency maintenance and differentiation[12,13,51]. These studies have relied on genetic YAP perturbation strategies and single time point analysis of YAP levels. Our identified concentration-dependent and dynamic decoding modes were made possible by our direct manipulation and measurement of YAP in living cells in conjunction with dynamic readouts of YAP target transcriptional activation. While optogenetic systems for the control of YAP in other model systems have previously been reported[52–55], we developed a tool box for the quantitative and dynamic control of YAP in embryonic stem cells. Furthermore, by comparing naive (2i+LIF) and differentiating cells, we demonstrate important differences in mESC signaling

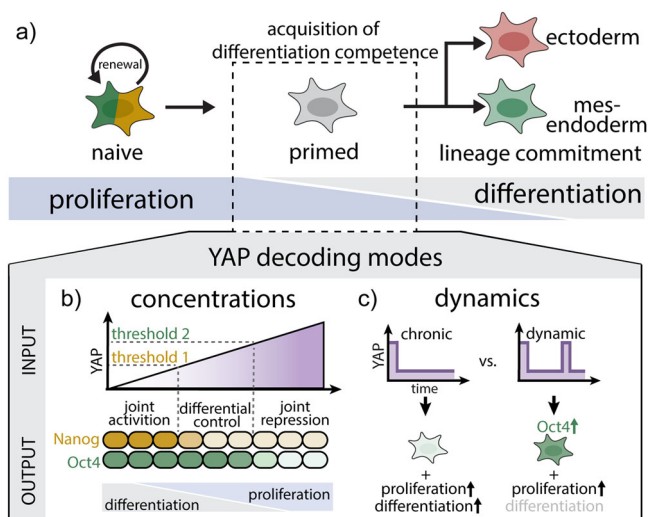

**Fig. 7 | Differential control of pluripotency factors and cellular decision-making through YAP levels and dynamics.** YAP concentrations and dynamics establish differential control strategies for pluripotency factors and developmental decision-making (proliferation, differentiation). **a** Top: Following pluripotency exit of naive mESCs, cells acquire signaling competence to interpret differentiation cues and transition to a primed state before committing to the ectodermal and mesendodermal lineages. The process is accompanied by progressive loss of proliferative potential. Bottom: Differential control strategies of cell behavior during the transition from pluripotency to lineage commitment through YAP levels and dynamics. **b** Decoding YAP steady-state concentrations through different sensitivity thresholds establish windows for the joint activation (low YAP) and repression (high YAP) or differential control (intermediate YAP) of Oct4 and Nanog. Across the YAP concentration range, steady-state levels have opposing effects on differentiation (YAP low) and proliferation (YAP high). **c** YAP dynamics enable the differential control of Oct4, cell differentiation, and proliferation. The different decoding modes depicted in **b**, **c** provide a means for the complex regulatory requirements of the pluripotency factors and cellular differentiation vs. proliferation.

competence to YAP inputs. Under naive culture conditions, Oct4 and Nanog are insensitive to a wide range of steady-state YAP concentrations, suggesting that the remodeling of the pluripotency network during pluripotency exit imparts YAP sensitivity. Our results not only address the long-standing debate on the function of YAP but also provide strategies to rationally correct YAP signaling in disease states through targeting YAP dynamics.

How genes distinguish between different steady-state concentrations and dynamics remains an exciting open question. We identify that the TEAD/YAP binding sites of the Oct4 regulatory region establish the negative cooperativity of the Oct4 threshold system to steady-state YAP levels. Similar mechanisms could enable the different repression thresholds observed for Oct4 and Nanog.

Why are YAP effectors like Oct4 and cellular decisions such as proliferation sensitive to a particular frequency of pulsatile YAP dynamics? We identify an adaptive circuit that generates a transient burst of Oct4 transcription following an acute drop of YAP concentrations. The adaptive response operates on the hour time scale, shows a significant delay between YAP change and transcription onset (~45 min), and operates by transient modulation of transcription burst frequency that resets to baseline under continuous YAP export. How these response characteristics relate to the underlying molecular machinery acting at the Oct4 locus, and if the machinery driving proliferation makes use of a similar adaptive strategy to decode dynamics, remain important open questions.

While our study focuses on the decoding logic of YAP levels and dynamics, our finding that differentiating mESC exhibit sporadic YAP pulses poses exciting new questions about information encoding

through YAP: What information is encoded in YAP levels and dynamics? Cellular differentiation is accompanied by substantial changes in cellular metabolism and morphology—for example from cell–cell or cell–ECM adhesion, alterations of cell density, the establishment of cell polarity, and changes in the mechanical properties of the environment (e.g., ECM deposition, hydraulic pressure of the blastocyst lumen). Given that YAP is mechanoresponsive, it is tempting to speculate that different cellular mechanical inputs may be encoded in YAP dynamics or concentrations. Coupling reporters for YAP dynamics with different mechanical inputs could help address this question.

The use of signaling dynamics to encode information is widespread in biology. Many other transcriptional regulators such as p53, NFkB or Erk make use of dynamic communication codes to pair upstream inputs to physiologically-relevant responses[21,45,56]. What is the advantage of dynamic modes as compared to steady-state cell signaling? Cells live in noisy environments where steady-state concentrations are subject to significant fluctuations. Dynamic readouts, such as the change sensor identified for Oct4, provide robustness to these fluctuations and are only engaged when specific temporal patterns are induced[57–59]. Similarly, behavioral coordination is crucial for proper formation of developmental shape and pattern, and temporal signals provide a means to synchronize cellular decision-making[60–63]. For example, Oct4-mediated induction of mesendoderm differentiation is most potently induced in G1 phase of the cell cycle[64]. Dynamic YAP signaling could synchronize Oct4 expression with the cell cycle of either individual cells or entire cell populations. Lastly, dynamic signaling decoders often act independent of absolute concentrations but make use of relative measures such as fold-changes[65–67]. This scale invariance significantly extends the dynamic range of signaling systems, thereby enabling control of cell signaling in more diverse environments than fixed concentration regimes would allow.

Together, our work emphasizes the central role of signaling dynamics as cellular communication code and paves the way for deploying synthetic YAP signaling inputs for tissue engineering and biomedical applications. Towards that goal, our light-gated YAP signaling toolbox provides a unique means to computationally interface with YAP signaling allowing to build more sophisticated feedback control systems.

## Methods
### mESC culture
E14 mESCs (gift from the Panning lab, UCSF) were maintained on gelatin coated dishes in 2i+LIF media, composed of a 1:1 mixture of DMEM/F12 (Thermo Fisher, 11320−033) and Neurobasal (Thermo Fisher, 21103−049) supplemented with N2 supplement (Thermo Fisher, 17502−048), B27 with retinoic acid (Thermo Fisher, 17504−044), 0.05% BSA (Thermo Fisher, 15260−037), 2 mM GlutaMax (Thermo Fisher, 35050−061), 150 μM 1-thioglycerol (Sigma, M6145), 1 μM PD03259010 (Selleckchem, 1036), 3 μM CHIR99021 (Selleckchem, S2924) and 10^6 U/L leukemia inhibitory factor (Peprotech, 250−02).

### CRISPR editing and cell line generation
For the generation of mESC reporter lines, we used the sgRNA/Cas9 dual expression plasmid pX330 (Addgene Plasmid 42230) and inserted sgRNA coding sequences targeting the YAP and Oct4 locus. Homology arm sequences for generation of knock-in donor vectors were amplified from E14 cDNA. pX330 and knock-in donor plasmids were introduced into mESCs by electroporation using the Neon Transfection System (Thermo Fisher Scientific, MPK10025). Cells were transfected with 400 ng pX330 plasmid and 600 ng donor plasmid per 150 000 cells and electroporated with the following settings: 1400 V, 10 ms pulse width, three pulses. Cells were recovered for 2 days in 2i+LIF media prior to clonal isolation.

For the generation of two different YAP KO lines, the YAP start sequence was targeted using the guide sequences 5′- CGGCTGT TGCGCGGGCTCCA-3′ and 5′-ACCAGGTCGTGCACGTCCGC-3′. A single clone was isolated for each knock-out line and characterized by sequencing and Western Blot. The isolated clones had insertions near the guide targeting site resulting in a frame shift and premature stop codon. Both clones were used as KO background for expression of the LEXY-YAP or YAP constructs.

For the generation of the endogenous SNAP-YAP reporter line in the WT mESC background, the same guide as for one of the YAP KO lines was used (guide: 5′-CGGCTGTTGCGCGGGCTCCA-3′) and co-transfected with a SNAP-YAP donor plasmid for homologous recombination. We constructed a donor plasmid that inserted a SNAP-tag sequence upstream of the start of the YAP coding region using flanking homology arms of ~800 bp. The SNAP-YAP line was verified to be a homozygous knock-in by sequencing. For simultaneous imaging of endogenous SNAP-YAP and GFP, a GFP expression cassette was additionally introduced into the SNAP-YAP line using the ePiggyBac transposase knock-in vector. Low GFP expressing cells were selected using FACS.

For knock-in of the 24×MS2 array into the Oct4 locus in WT and YAP KO mESCs, we constructed a donor plasmid that inserted a 24×MS2 cassette, followed by a start codon, puromycin coding sequence and P2A sequence, directly upstream of the first exon of the Oct4 locus using homology arms of ~500 bp. The sgRNA used for knock-in of this cassette targeted the Oct4 5′UTR (5′-TTTCCAC-CAGGCCCCCGGCT-3′). The isolated clones (WT or YAP KO background) harbor a single allele with the MS2 cassette. To detect the MS2 array, we introduced the MS2 coat protein fused to two copies of the Halo-tag using the ePiggyBac transposase knock-in vector[68] to generate stable lines. PiggyBac knock-in of the MS2 coat protein was also used to express a nuclear marker for the endogenous SNAP-YAP cell line. All MS2 coat protein expressing lines were sorted for low MS2 coat protein expression by FACS.

For expression of the Oct4 WT and TEAD mutant reporter, the reporters were randomly integrated into the YAP KO line using the ePiggyBac transposase knock-in vector and positive cells were selected using FACS.

## Cloning of doxycycline inducible LEXY-YAP, YAP, SNAP and TEAD reporter constructs

The YAP sequence used for all expression constructs was amplified from E14 mESC cDNA and represents the mouse isoform that lacks exon 6. The iLEXYi sequence[30,31] was generated by point mutation (V416I) of the LEXY sequence from the NLS-mCherry-LEXY plasmid (Addgene 72655). An NLS-SNAP-iLEXYi cassette was fused to the N-terminus of YAP and expressed under a doxycycline inducible cassette. As non-light responsive controls, we expressed the same construct but lacking the iLEXYi sequence. The non-light responsive YAP control for the Oct4-MS2 live imaging experiments contained the additional NLS at the N-terminus of the SNAP-tag (Fig. 4d). The non-light responsive YAP control used in all other experiments lacked the additional NLS sequence. A doxycycline inducible NLS-SNAP vector was cloned and used as transfection control of WT mESCs for analysis of lineage specification in Fig. 6. The Oct4 WT and TEAD mutant reporter were cloned from the Addgene plasmid #60527[38]. The four TEAD sites predicted by the JASPAR database[69] and reported by Lian et al. [12] were deleted using primer mutation. The reporter was cloned into the ePiggyBac transposase knock-in vector.

## DNA FISH

The Oct4 DNA FISH probe was generated using the BioPrime DNA labeling system (Invitrogen, 18094-011) and dUTP-Alexa488 labeled nucleotides (ChromaTide™ Alexa Fluor™ 488-5-dUTP, C11397)

according to manufacturer's instructions. As DNA template, we used the BAC vector RP24-241N15 (BACPAC Genomics) containing part of the Oct4 locus. For DNA FISH staining, etched grid coverslips (Bellco Biotechnology, 1916-92525) were coated with 10 μg/ml natural mouse laminin (Thermo Fisher, 23017015) for ~6 h at 37 C, and cells were seeded in 2i+LIF media. Post 24 h of seeding, cells were stained with Halo-tag ligand JF549 (see below) and fixed with 4% paraformaldehyde for 10 min at room temperature. Because the JF549 staining was lost during the DNA FISH staining procedure, we imaged the fixed Oct4-MS2 (JF549) staining prior processing for DNA FISH and realigned cells with the DNA FISH staining using the grid on the etched slides. For DNA FISH, cells were washed with 0.5% Tween/PBS (Fisher Scientific, BP337-500) for 10 min and incubated overnight in 70% ethanol (Sigma Aldrich, E7023) at 4 C. Slides were consecutively incubated in 2× saline sodium citrate (SSC; Norgen Biotek Corp., 28157) for 10 min, 0.1 N HCl (Fisher Scientific, AC423795000) for 5 min and PBS for 5 min on ice. Slides were air dried and incubated in hybridization solution (10× SSC, 5 mg/ml BSA, 25% dextran sulfate; New England Biolabs, B9000; Sigma Aldrich, D4911) at 80 C for 7.5 min. Slides were dehydrated by sequential incubation in 70%, 80% and 100% ethanol for 5 min each. Slides were air dried and incubated with the Oct4 DNA FISH probe in 10× SSC, 5 mg/ml BSA, 25% dextran sulfate, 50% formamide (Sigma Aldrich, 11814320001) for 5 min at 80 C followed by incubation at 37 C overnight. Slides were washed in wash buffer (55% formamide, 2×SSC, 0.1% NP-40) at 42 C for 10 min, washed with PBS and mounted.

## Transient transfection of mESCs with LEXY-YAP, YAP and SNAP vectors

For all experiments using the doxycycline inducible YAP or LEXY-YAP system, YAP KO mESCs were transiently transfected with doxycycline inducible LEXY-YAP and non-light responsive YAP vectors. Transfection of WT mESCs with an inducible NLS-SNAP vector served as control in Fig. 6a, b. To transfect mESCs, 5 × 10^6 mESCs were electroporated with 6.6 μg plasmid using the Neon Transfection System (Thermo Fisher Scientific, MPK10025). Neon settings for the electroporation were as follows: 1400 V, 10 ms pulse width, three pulses. Following electroporation, cells were seeded in 2i+LIF media supplemented with 1% ES-qualified FBS and 100 ng/ml doxycycline or doxycycline concentrations as indicated. 24 h post electroporation, cells were stained with SNAP-tag ligand JF646 (see below) prior FACS sorting for positive cells. Positive cells were seeded for spontaneous differentiation as described in the "Spontaneous differentiation" section (see below).

## Directed differentiation

For directed differentiation of endogenously tagged SNAP-YAP mESCs into the mesoderm lineage, 96-well glass bottom dishes (Cellvis, P96-1.5H-N) were coated with 10 μg/ml natural mouse laminin (Thermo Fisher, 23017015) for ~6 h at 37 C, and cells were seeded at 10 000 cells per well in 2i+LIF media. 12 h post seeding, cells were washed three times with DMEM/0.05% BSA (Thermo Fisher Scientific, 11995-073; Gibco, 5260037) and cultured in differentiation media composed of a 1:1 mixture of DMEM/F12 (Thermo Fisher Scientific, 11320−033) and Neurobasal (Thermo Fisher, 21103−049) supplemented with N2 supplement (Thermo Fisher, 17502−048), B27 without retinoid acid (Thermo Fisher, 12587010), 0.05% BSA (Thermo Fisher, 15260−037), 2 mM GlutaMax (Thermo Fisher, 35050−061), 150 μM 1-thioglycerol (Sigma, M6145) and 3 μM CHIR99021 (Selleckchem, S2924). Media was changed daily. Cells were imaged at different time points post differentiation or fixed 5 days post differentiation start for IF staining of lineage markers.

For directed differentiation of endogenously tagged SNAP-YAP mESCs into the ectoderm lineage, cells were cultured in 2i+LIF SFES media in presence of 15% ES-qualified FBS (Gibco, 16141079) for 24 h. 96-well glass bottom dishes were coated with 10 μg/ml natural mouse

laminin (Thermo Fisher, 23017015) for ~6 h at 37 C, and cells were seeded at 5000 cells per well in 2i+LIF media supplemented with 15% ES-qualified FBS. 12 h post seeding, cells were washed three times with DMEM/0.05% BSA and cultured in differentiation media composed of a 1:1 mixture of DMEM/F12 (Thermo Fisher Scientific, 11320–033) and Neurobasal (Thermo Fisher, 21103–049) supplemented with N2 supplement (Thermo Fisher, 17502–048), B27 with retinoid acid (Thermo Fisher, 17504044), 0.05% BSA (Thermo Fisher, 15260–037), 2 mM GlutaMax (Thermo Fisher, 35050–061), 150 µM 1-thioglycerol (Sigma, M6145). Post 24 h of differentiation start, 1 µM retinoic acid was added to the media. Media was changed daily. Cells were imaged at 1.5d post differentiation or fixed 5 days post differentiation start for IF staining.

## Spontaneous differentiation

Media conditions: for spontaneous differentiation, 2i+LIF media was removed, and cells were washed three times with DMEM/0.05% BSA prior to differentiation start. Two different spontaneous differentiation media were used favoring differentiation into the mesendodermal (FBS-based) or ectodermal (N2B27-based) lineage. FBS-based spontaneous differentiation media was composed of DMEM high glucose (Thermo Fisher Scientific, 11995-073), 15% ES-qualified FBS (Thermo Fisher, 16141079), 2mM L-Glutamine (Gibco, 35050061), 0.1 mM non-essential amino acids (Gibco, 11140-050), 150 uM thioglycerol (Sigma Aldrich, M6145), supplemented with doxycycline (Sigma Aldrich, D9891; see below). N2B27-based spontaneous differentiation media was composed of a 1:1 mixture of DMEM/F12 (Thermo Fisher Scientific, 11320–033) and Neurobasal (Thermo Fisher, 21103–049) supplemented with N2 supplement (Thermo Fisher, 17502–048), B27 with retinoid acid (Thermo Fisher, 17504044), 0.05% BSA (Thermo Fisher, 15260–037), 2 mM GlutaMax (Thermo Fisher, 35050–061), 150 µM 1-thioglycerol (Sigma, M6145). Doxycycline was added at 100 ng/ml to all media, except for the doxycycline titration experiments where doxycycline was added at the indicated concentrations. Media was replaced daily.

Seeding and culture conditions: for IF analysis of Oct4 and Nanog protein levels and Oct4-MS2 transcription, 64kPa PDMS 96-well glass bottom dishes (Advanced Biomatrix, 5261) were coated with 10 µg/ml natural mouse laminin (Thermo Fisher, 23017015) for ~6 h at 37 C, and cells were seeded at 15 000 cells per well. The PDMS substrates were chosen to ensure a defined permissive mechanical environment as previously established for YAP's role in cellular reprogramming[70]. Cells were fixed at 1.5d post differentiation start for IF analysis of Oct4 and Nanog protein levels or imaged between 1 and 1.5d post differentiation start for analysis of MS2-Oct4 transcription.

For analysis of lineage specification and proliferation, 96-well glass bottom dishes (Cellvis, P96-1.5H-N) were coated with 10 µg/ml natural mouse laminin (Thermo Fisher, 23017015) for ~6 h at 37 C, cells were seeded at 1000 cells per well and fixed at 5 days post differentiation start.

For the analysis of the early differentiation marker Otx2 and proliferation upon light illumination, 96-well glass bottom dishes (Cellvis, P96-1.5H-N) were coated with 10 µg/ml natural mouse laminin (Thermo Fisher, 23017015) for ~6 h at 37 C, cells were seeded at 1000 cells per well and cells were fixed at 2d post differentiation start.

## Mapping of Oct4 levels in the inducible YAP system to the endogenous Oct4 range

To map Oct4 levels in our inducible systems to native conditions during differentiation, WT cells were differentiated at 1000 cells per well and compared to LEXY-YAP expressing cells seeded at 15000 cells per well and fixed 1.5d post differentiation. All cells were grown on laminin coated 64kPa PDMS glass bottom dishes as described above and fixed for Oct4 IF staining. LEXY-YAP cells were additionally stained for SNAP-tag ligand JF646 prior to fixation to map Oct4 to LEXY-YAP levels.

## Mapping of endogenous YAP range to LEXY-YAP expression levels

YAP IF staining was used to compare the endogenous YAP level range of WT mESCs to the expression range of our LEXY-YAP tool. Since fixation of the JF646 SNAP-tag ligand affects its intensity, we first live imaged the cells to detect their LEXY-YAP JF646 signal. Then, the LEXY-YAP cells and WT mESCs (grown under the same conditions) were PFA fixed and YAP levels were determined by YAP immunofluorescence. The median LEXY-YAP levels of the live imaging LEXY-YAP (LEXY-YAP$_{JF646}$) and the LEXY-YAP immunostaining (LEXY-YAP$_{IF}$) cell populations were used to obtain a conversion factor that maps the WT YAP levels (WT$_{IF}$) to the live imaging intensities using LEXY-YAP$_{JF646}$/LEXY-YAP$_{IF}$*WT$_{IF}$.

## Light-gated YAP dynamics for analysis of Oct4, Nanog, Otx2 protein levels and cell proliferation

24 h post spontaneous differentiation start, cells were exposed to 470 nm LED light illumination patterns in a tissue culture incubator (37 C, 5% CO2) using the optoPlate-96[71]. For all illumination conditions, cells expressing non-light responsive SNAP-YAP served as controls. Oct4 and Nanog levels were determined after a 12 h illumination period (at 36 h post differentiation start), while Otx2 and cell numbers were analyzed after a 24 h illumination phase (at 48 h post differentiation start). For experiments quantifying Oct4 and Nanog as a function of YAP levels, LEXY-YAP and YAP was detected using the JF646 SNAP-tag ligand (see below) and stained at 11 h post illumination start under continuous light illumination. For experiments quantifying Otx2 levels as a function of YAP or LEXY-YAP, YAP was detected by YAP IF staining. For all experiments, cells were recovered for 20 min in the dark following the illumination phase to reset LEXY-YAP localization to the dark state. This allowed us to compare cells based on their LEXY-YAP and YAP expression levels between all illumination conditions. Cells were fixed in the dark with 4% PFA (Electron Microscopy Sciences, 15714-S) for 30 min and washed twice with PBS (Life Technologies, 14040133) prior to IF staining for Oct4 and Nanog or Otx2 and YAP (see section "IF staining of mESCs").

## Light-dose titration of the LEXY-YAP tool

To establish the light titration curve, LEXY-YAP transfected cells were differentiated in spontaneous differentiation media at 15 000 cells/ well for 1.5d on 96-well glass bottom plates (Cellvis, P96-1.5H-N) and illuminated with different light intensities for 20 min. Cells were PFA fixed under continuous illumination and stained for YAP by IF to quantify YAP export in reference to a dark control.

## Staining with SNAP and Halo-tag ligands

Cells were incubated with 10 nM SNAP-tag ligand JF646 or Halo-tag ligand JF549[72] for 30 min in their respective culture media, washed 3 × 5 min with DMEM/0.05%BSA (Thermo Fisher Scientific, 11995-073; Gibco, 5260037) and incubated for 1 h in culture media prior to further processing.

## IF staining of mESCs

Fixed cells were permeabilized with 0.05% TritonX-100/0.075% Sodium dodecyl sulfate (Fisher Scientific, BP151-100; Sigma Aldrich, 436143) for 20 min and blocked with 10% normal goat serum (Abcam, ab7481; for staining with Oct4 and Nanog) or 5% BSA (Sigma Aldrich, A8806; for staining with Sox1, Tbra, FoxA2, Otx2, YAP) for 1 h. Cells were incubated with 1:200 primary antibody against Sox1 (Cell Signaling Technology, 4194 S), Tbra (RD Systems, AF2085), FoxA2 (Santa Cruz, sc-374376), Oct4 (Santa Cruz, sc-5279) Nanog (Cell Signaling Technology, 8822 S), YAP (Cell Signaling Technology, 14074) or Otx2 (R&D Systems, AF1979) in blocking buffer for 2 h at room temperature, washed three times with 0.01% TritonX-100 (Fisher Scientific, BP151-100) and incubated with Alexa-488, −568 or −647 conjugated

secondary antibody (1:1000, Thermo Fisher Scientific) and NucBlue (Thermo Fisher Scientific, R37605) in blocking buffer for 1 h at room temperature. Cells were washed 3 × 15 min with 0.01% TritonX-100 and incubated in PBS for imaging.

## Imaging of IF stained samples

IF stainings were imaged on a Nikon Eclipse Ti inverted confocal microscope (Nikon) equipped with a CSU-W1 Yokogawa spinning disk (Andor), an iXon Ultra EMCCD camera (Andor), and 405, 440, 488, and 561-nm laser lines using a 40× Plan Apo TIRF 0.95 NA air objective (Nikon). Wells were imaged as non-overlapping 10 × 10 or 8 × 8 image grids. Pixel size was 0.325 µm.

## Live imaging of endogenous YAP dynamics

For live imaging of endogenous SNAP-YAP upon directed differentiation into the mesoderm and ectoderm lineages, with or without simultaneous imaging of GFP, cells were stained with SNAP-tag ligand JF646 and Halo-tag ligand JF549. Cells were imaged on an environmentally controlled (37 C, 7% CO$_2$) Nikon Eclipse Ti inverted confocal microscope (Nikon) with the same specifications as described for the imaging of IF stained samples (see above). Cells were imaged with a 60× Apo TIRF 1.49 NA oil objective (Nikon) at 10 min intervals for a total of 16 h. Pixel size was 0.216 µm.

## Live imaging of transcription and optogenetic stimulation

To reduce photobleaching for live imaging of transcription, media was exchange to media without phenol red (R&D Systems, M18650) and supplemented with 50 µg/mL ascorbic acid (Sigma Aldrich, A4544) and 1:100 Prolong Live Antifade Reagent (Thermo Fisher, P36975) 30 min prior to imaging start. Imaging was performed on a confocal Nikon Ti microscope (Nikon) equipped with a CSU-22 spinning disk confocal unit, an Evolve Delta EMCCD camera (Photometrics) and an environmentally controlled stage incubator (37 C, 5% CO$_2$).

For imaging the Oct4-MS2 reporter in WT and YAP KO lines, cells were imaged with a Plan Apo VC 60×/1.4 NA oil objective (Nikon, pixel size 154 nm). Images were acquired as z-stacks with planes spaced 200 nm apart, covering a total of 8–10 µm in z (= 41-51 z slices) and acquired every 30 sec for a total of 1 h.

For simultaneous optogenetic control of nuclear LEXY-YAP levels and imaging of the MS2 transcriptional reporter, cells were imaged with a Plan Fluor 40×/1.3 NA oil objective (Nikon, pixel size 228 nm) using a dual band pass red and far-red emission filter that blocks blue light. 470 nm LED light for optogenetic light stimulation was applied using the optoPlate-96[71]. The plate was mounted on top of the stage such that the temperature and CO$_2$ control was maintained. Light was applied pulsed with a 1 sec ON/1 s OFF duty cycle. The MS2 signal was acquired as z-stack with planes spaced 250 nm apart, covering a total of 17 µm in z (= 71 z slices) and acquired every 1.5 min. The LEXY-YAP or YAP signal was captured as single slice in the center of the stack and acquired every 22.5 min. The LEXY-YAP and YAP control cells were imaged in adjacent wells in the same run. For all MS2 and LEXY-YAP imaging experiments, the laser output was manually measured and adjusted for every experiment to quantitatively compare LEXY-YAP, YAP and MS2 intensities between experiments.

## Denoising of microscopy images

Images from time lapse microscopy of the MS2 reporter and endogenous YAP were denoised using NDSafir[73,74].

## Quantification of endogenous YAP levels from live imaging time courses

Denoised images were background subtracted using a dark field image and flat field corrected. Nuclei were segmented based on the nuclear marker (MCP) using the StarDist detector[75] and tracked using the overlap tracker in the Fiji Trackmate plugin[76]. Small objects were filtered out, debris or dead cells were excluded manually and only tracks with track length >= 10 h were kept. Trackmate quantifications of the median nuclear YAP intensity were imported into Python (version 3.8.5) for further processing. The first and last three frames of each cell trace were removed to ensure cells were not entering or exiting mitosis. For each nucleus both the nuclear marker (MCP) and the YAP signal were normalized to the average MCP or YAP signal computed from all nuclei of the imaging run, to correct for bleaching and fluorescent drift. Each MCP or YAP trace was then normalized to its minimum value. Cells with substantial changes of the nuclear MCP marker were excluded as this indicated tracking error or mitotic cells. To this end, the rolling standard deviation of the normalized MCP trajectories was computed (window size: 10 time points), and nuclei with MCP trajectories containing standard deviations > 0.05 were excluded from the analysis. To analyze YAP dynamics, the normalized single cell YAP trajectories were smoothed using a rolling mean (window size: 2 time points) and peak detection was performed using the find_peaks function from the python signal module, with peak prominence=0.25 and height > =1.3. The find_peaks function finds all local maxima by simple comparison of neighboring values and further filters traces for the absolute peak height. From this, we computed the percentage of cells pulsing (= cell that show at least one YAP pulse within a minimum of 10 h) as well as the pulse duration and amplitude. The same analysis pipeline was used to determine the false positive rate using a GFP control or fixed SNAP-YAP cells.

## Quantification of nuclear LEXY-YAP levels from live imaging time courses

The mean nuclear LEXY-YAP intensities were manually quantified from time lapse movies by drawing a rectangle in the nucleus using Fiji[77].

## Quantification of Nanog and Oct4 from microscopy images and sigmoidal curve fit

Nuclei were segmented based on the NucBlue (Hoechst) staining using the Fiji StarDist plugin[75]. Small objects were filtered out and the mean nuclear Oct4, Nanog and LEXY-YAP or YAP intensities were quantified for each nucleus. For establishing the sigmoidal curves in Fig. 2c, the LEXY-YAP and YAP intensities were scaled using the dark control to compare between experiments. To this end, for each experimental repeat, we plotted the Nanog and Oct4 intensities as a function of LEXY-YAP for the dark condition and fit a sigmoidal function to determine the IC50s for Oct4 and Nanog using the GraphPad software. We then determined the center of the Nanog and Oct4 IC50s for each experimental repeat and used that value to scale the LEXY-YAP and YAP intensities for every experimental repeat. Following scaling of the LEXY-YAP intensities, the Oct4 and Nanog signal of each experiment was min-max normalized using the bottom and top plateau of each sigmoidal curve fit as min and max, respectively. Note that the lowest bin containing cells with no detectable LEXY-YAP/YAP expression was excluded from the dataset and analysis. These cells lost expression during the experiment making them uninterpretable.

For quantification of the nuclear Oct4 and Nanog intensities upon light-gated oscillatory YAP inputs, we only considered cells with LEXY-YAP levels that were sufficient to repress Oct4 in the dark control (>IC50 or >IC5 of the dark control, as indicated in text or legends). To this end, we plotted the Oct4 intensities as a function of LEXY-YAP intensities for the dark control, fit a sigmoidal function and determined the IC50 using GraphPad. The median Oct4 and Nanog protein levels of all cells with LEXY-YAP or YAP levels higher than the dark well Oct4 IC50 or IC5 was determined for each illumination condition. Data was normalized to the Oct4 levels in low expressing LEXY-YAP cells of the dark control and the difference of the normalized data between the experimental and dark condition was quantified (ΔOct4).

## Quantification of Sox1, Tbra and FoxA2 positive cells, nuclear Otx2 intensities and cell numbers

Nuclei were segmented using the NucBlue channel as described for the quantification for nuclear Nanog and Oct4 from IF stainings (see above). Segmentation masks were used to quantify the median nuclear lineage markers for each nucleus. To establish cut-off values for the identification of Sox1, Tbra and FoxA2 positive cells, we made use of opposing fates to determine background intensities, assuming that cells can only acquire one germ layer fate. For example, to determine the cut-off value for Tbra positive cells, we manually gated cells with the highest Sox1 levels and determined the median nuclear intensity and standard deviation for the opposing marker (Tbra) in that gate. The median + 2* SD of that value was then used as cut-off value to define Tbra positive cells.

We used the same strategy to establish cut-off values for the quantification of Tbra, FoxA2 and Sox1 positive cells for undirected differentiation experiments. Here, we determined the FoxA2 and Tbra background staining from Sox1 high cells as well as the Sox1 background intensities from Tbra high cells.

To quantify Otx2 intensities and cell numbers as a function of LEXY-YAP or YAP levels, the LEXY-YAP and YAP levels were scaled between experimental repeats. For every experiment, the Otx2 intensity was plotted as a function of LEXY-YAP or YAP levels for the dark well control and a line was fit to determine the IC50 using the GraphPad software. LEXY-YAP and YAP values were then scaled to the average IC50 of all experimental repeats. Experiments were binned for LEXY-YAP or YAP values and Otx2 intensities were normalized to median Otx2 intensity of the lowest LEXY-YAP or YAP bin. The average Otx2 intensity was quantified for every LEXY-YAP or YAP bin.

Cell numbers were quantified from the same LEXY-YAP or YAP-scaled datasets as the Otx2 quantifications (see above). The proliferative response upon light illumination showed a dependency on LEXY-YAP levels, requiring intermediate LEXY-YAP levels for a maximum response. Cells numbers were therefore quantified only in cells expressing LEXY-YAP or YAP with at least 5000 a.u. average nuclear LEXY-YAP or YAP intensity (see Supplementary Fig. 10c dashed line).

## Quantification of MS2 spot intensities from time lapse movies

3D time-lapse MS2 image stacks were converted into 2D images by maximum Z projection using Fiji. The AI segmentation algorithm form the NIS.ai suite of the NIS-Elements software (Nikon) was used for initial MS2 spot detection and nuclear segmentations from these projections. The segmented images were fed into a Python analysis pipeline to verify and quantify the MS2 spots, assign them to nuclei, track them over time and quantify nuclear YAP and MS2 coat protein intensities.

To track individual nuclei, the Python package trackpy (version 0.4.2) was used. Nuclei with track length shorter than the total imaging duration, that left the field of view (touch the image border) or that were dividing, were excluded from the analysis. Mean nuclear MCP, LEXY-YAP or YAP intensities were quantified and background subtracted. For the MCP signal, the nuclear intensity was quantified on the center slice of the z-stack.

To quantify MS2 spot intensities, spots identified by the Nikon AI were further verified by a 2D gaussian fit on the maximum projection, followed by a 1D gaussian fit along the z-dimension of the image stack. Spots identified to be true MS2 signal were background subtracted using the local mean MCP signal intensity in vicinity to the sport. The sum pixel intensities within the identified spot were quantified from the z-stack. The MS2 signal was corrected for cell-to-cell differences in MS2 coat protein expression levels. To this end, the average spot intensity per nucleus was plotted against the MS2 coat protein levels and a second order polynomial was fit to the data using GraphPad (see Supplementary Fig. 8d). Only nuclei with non-saturating MS2 coat protein levels (50–200 a.u., see gray shading in Supplementary Fig. 8d)

were used for quantifications and the MS2 spot intensities were corrected for MS2 coat protein expression levels using the fit. If in rare events nuclei contained more than one MS2 spot (e.g., after S-phase), the intensity of the spots was summed up. Movies were manually inspected for fluorescent particles/dirt that were falsely identified as MS2 signal and identified nuclei (rare cases) were excluded from the analysis. For time frames with no detectable MS2 signal the intensity was set to zero.

For analysis of the Oct4-MS2 response to light-gated nuclear LEXY-YAP export, we only considered cells with LEXY-YAP or YAP levels within the endogenous YAP level range (see gray shading in Fig. 4c), which we define as levels above background and below the 95% percentile of the endogenous YAP level distribution. We further distinguished cells based on their dark state MS2 transcriptional activity (first 45 min of the time course). Only nuclei with low to moderate dark phase transcriptional activity were used for analysis. Cells that were transcriptionally quiescent in the dark phase showed no detectable difference between the control and light-gated LEXY-YAP export condition (see Supplementary Fig. 8e).

## Software

Data was collected and analyzed using Micro-Manager version 2.0 gamma, Arduino IDE 1.8.13, Matlab 2020a, Python 3.8.5, Fiji(imageJ) 2.14.0, GraphPad Prism 9.1.0, and Nikon NIS-Elements AR.21.01.

## Statistics

Except for burst parameter inference (Fig. 5), all statistical analysis and curve fits were performed using GraphPad (GraphPad software, Inc). Details can be found in the legend of each figure. N represents the number of independent experiments. $P$-values < 0.05 were considered statistically significant. Data was assumed to be normally distributed. t tests were not corrected for multiple comparisons. Statistical analysis for burst parameter inference methods are described below.

## Burst parameter inference methods

**cpHMM**. The burst parameter trends shown in Fig. 5 were obtained using cpHMM, a computational method that employs compound-state Hidden Markov Models to infer promoter state dynamics and burst parameter values (frequency, duration, and amplitude) from populations of single-cell traces. See[49] for details regarding the method's implementation. Briefly, transcriptional traces were divided into inference groups according to either KO/WT status (Fig. 5b), average nuclear YAP concentration (Fig. 5c), or according to time relative to optogenetic perturbation (Fig. 5d). Parameter estimates for each inference group were estimated by taking the average across no fewer than 16 separate bootstrap samples. Each bootstrap sample contained at least 1000 time points. Outlier bootstrap results were excluded using Matlab's built-in "rmoutliers" function, which defines outliers as any value that is more than three scaled median absolute deviations from the population median. Inference uncertainty was estimated by taking the standard deviation across these bootstrap replicates. We used a model with two burst states (OFF and ON), as illustrated in Fig. 5a.

**Estimating elongation times**. A key input parameter for cpHMM inference is the amount of time required for RNA Polymerase molecules to transcribe the reporter gene. We estimated this quantity for our Oct4 reporter by examining "low-to-high" transition events in our MS2 data. Specifically, we set a "high" threshold, $f_{high}$, for each inference group, defined as the 85th percentile of all observed fluorescent spot intensities and a "low" threshold defined as $f_{low} = 0.15 f_{high}$. We use these thresholds to filter for instances in our MS2 traces where the system transitions from low ($f \leq f_{low}$) to high ($f \geq f_{high}$) fluorescence levels. We then took the average across these events to obtain an averaged low-to-high event.

Intuitively, the time required for the system to transition from low to high fluorescence captures how long it takes to change from having an empty (or nearly empty) reporter gene–low fluorescence state–to a full (or nearly full) reporter gene–high fluorescence state. Accordingly, we used Matlab's built-in "findchangepts" function to estimate the number of time steps required to transition from basal fluorescence levels to saturating levels for each reporter gene, and defined this quantity as the elongation time. For the KO vs. WT experiment depicted in Fig. 5b, which had an experimental time resolution of 30 s, this procedure produced elongation time estimates of 8 steps (240 s) for Oct4. For the optogenetic results depicted in Fig. 5d, which had a time resolution of 90 s, we obtained an elongation time estimate of 3 steps (270 s) for Oct4.

**Estimating p-Values.** We used bootstrap cpHMM inference replicates to calculate p-Values for the results shown in Fig. 5b–d. To illustrate our approach, we describe the procedure in detail for the steady-state Oct4 response to YAP levels (Fig. 5c). The same approach was used to calculate p-Values for the remaining results in Fig. 5.

Figure 5c shows burst parameter results for Oct4 loci exposed to low, medium, and high YAP concentrations. For each burst parameter, the trend is normalized by the "high" YAP value, such that the results report on the fold difference in burst frequency, duration, across YAP levels. Consider the fold change in burst frequency for intermediate ("mid") YAP levels. On average, the burst frequency increases by a factor of about 1.4 relative to high YAP levels. In this case then, we wish to establish how confident we can be in our finding that the fold change is significantly greater than 1.

To do this, we make use of our bootstrap replicates. Once outliers are removed, we have $N_{high}$ inference bootstraps for the high condition and $N_{mid}$ inference bootstraps for the intermediate condition. We then calculate the fold increase for all possible combinations of mid and high bootstrap results, leading to $N_{fold} = N_{high} \times N_{mid}$ distinct bootstrap estimates of the fold increase. Our p-value is then simply defined as the fraction of these $N_{fold}$ estimates that are found to be *less than or equal to* 1; i.e., the fraction of replicates for which the observed fold increase does not hold. If the trend is significant, this will occur only rarely, whereas this will occur frequently if a trend is small relative to its uncertainty. In this case, we find that about 20% of all bootstrap fold change values are less than or equal to 1. This equates to a p value of 0.2, which implies that the result is not significant at the 10%, 1%, or 0.1% levels. This same bootstrap-based approach is used to assess significance levels for all results shown in Fig. 5.

### Reporting summary
Further information on research design is available in the Nature Portfolio Reporting Summary linked to this article.

## Data availability
Source data are provided with this paper.

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

## Acknowledgements

We thank Jefferey Alexander for experimental advice and all members of the Weiner lab for their support. We are grateful for help with live imaging experiments from the Center for Advanced Light Microscopy at UCSF. The LEXY plasmid (Addgene plasmid #72655) was a gift from the Di Ventura and Eils labs. The Janelia fluor dyes were kindly provided by Luke Lavis. Funding: O.D.W. was supported by an NIH grant GM118167 and the National Science Foundation Center for Cellular Construction (DBI- 1548297). K.M. was supported by postdoctoral fellowships from the German Research Foundation (DFG, ME 5071) and the American Heart Association (20POST35180100). N.C.L. was supported by an NIH Genomics and Computational Biology training grant (5T32HG000047-18), the Howard Hughes Medical Institute, by DARPA award number N66001-20-2-4033, and by a Postdoctoral Fellowship from the Damon Runyon Cancer Research Foundation. L.J.B. was supported by an NIH grant R35GM138211. H.G.G. was supported by an NIH R01 Award (R01GM139913) and the Koret-UC Berkeley-Tel Aviv University Initiative in Computational Biology and Bioinformatics. H.G.G. is also a Chan Zuckerberg Biohub Investigator (Biohub – San Francisco).

## Author contributions

K.M. designed and performed experiments, analyzed, and interpreted data, drafted and edited the manuscript. N.C.L. analyzed and interpreted data and edited the manuscript. L.J.B. provided critical resources. H.G.G. interpreted data and edited the manuscript. O.D.W. supervised the study, acquired funding, helped design experiments and interpret data, and drafted and edited the manuscript.

## Competing interests

The authors declare no competing interests.
