## [Peer Review File · Nature Communications]

REVIEWER COMMENTS

Reviewer #1 (Remarks to the Author):

YAP is a canonical signaling regulator of important cellular decisions, such as during development. This manuscript explores the effects of YAP signaling dynamics on the expression of Oct4 and Nanog in mouse ES cells and the subsequent impact on cell differentiation. The questions are interesting, interesting novel tools were developed (the optogenetic YAP), and the experimental design and analysis were almost always well designed and executed. The relationship between YAP inputs and dynamic Oct4 responses are particularly interesting. That said, there are some significant questions, particularly about the interpretation.

The choice of time points used in the study sometimes has an unclear rationale. SNAP-YAP was analyzed 1.5 days after differentiation as a “a generally permissive time window for differentiation cues in mESCs.” This is a single time point, and it’s not clear whether cells at this time point are in the act of integrating YAP signal to influence the differentiation decisions later investigated in this study, or whether this is an arbitrary time point where the YAP dynamics are not relevant to the biology later studied. They then investigated the effect of their inducible YAP system (which is a clever system) on Oct4/Nanog expression at 2 days. Why use a different time point? Why use a single time point, when differentiation is a dynamic process?

The manuscript states “In response to an acute chronic input, adaptive signaling circuits transiently respond to the change in signal input but then reset to the initial baseline under sustained activation of the input.” “Acute chronic” is seemingly a contradictory phrase. In addition, the left panel of 3E shows a transient rather than a sustained activation of the input. This leads to some more general confusion about the proposed behavior of the system. Fig. 2C proposes to show that steady state nuclear levels of YAP drive different steady state levels of Oct4/Nanog. However, from the supplementary info, the experiment appears to involve spontaneously differentiating cells for 24 hours followed by illumination for 12 hours and measurement of Oct4/Nanog proteins at this 36 hour time point. Furthermore, Fig. 4 shows that a step increase in illumination (i.e. switching on the light) results in a pulse of Oct4 expression. In other words, constant illumination in Fig. 4 results in pulsatile Oct4 expression, all in the context of a cell that is undergoing spontaneous differentiation. How exactly is Fig. 2 then at steady state? Are the different Oct4 levels in Fig. 2C the result of the integral of different Oct4 transcriptional pulse sizes that happened 12 hours before, subtracting the effects of Oct4 protein and mRNA degradation during this 12 hours? In this case, how could Oct4 levels be referred to as steady state?

This brings up a significant concern with the following, a central conclusion of the manuscript: “Our results reveal two different YAP decoding modes by Oct4 that are implemented at the gene regulatory level: steady-state YAP levels control Oct4 in a dose-dependent manner, while acute YAP changes induce

an adaptive transient response.” If a step increase to a new steady state of YAP led to a pulse of Oct4 transcription that was read out 12 hours later as a new Oct4 “steady state” (when in fact Oct4 protein and mRNA could be undergoing degradation to return back to the same original level, and thus Fig. 2c is a snap shot in time rather than a steady state), then the first half of the manuscript’s sentence could be a downstream, misinterpreted consequence of the second half of the sentence. Or is this reviewer missing something?

At the transcriptional level, a different question seemingly emerges. Different “steady state” (12 hour?) levels of YAP (Fig. 5C) were initially established as step changes in nuclear YAP. However, Oct4 transcriptional burst frequency transiently increases with a step change in YAP before returning to baseline in ~3 hours (Fig. 5D). So unless a step increase in YAP leads to an initial pulse and decay to baseline, followed in the next 9 hours (where the manuscript does not have experimental observations) in some increase to a higher steady state, these two outcomes are contradictory. That is, does a step change in YAP result in a long term increase in burst frequency (Fig. 5C) or a transient increase in burst frequency (Fig. 5D).

it appears that a step increase in YAP illumination

The manuscript states that “adaptive systems generate optimal responses at a specific input frequency that matches the resetting time.” It’s not clear what the word optimal means in this context. What is it optimal for?

Can the authors describe molecularly how YAP is regulating Oct4/Nanog? This question isn’t even broached until the end of page 6 of the manuscript, and the wording is somewhat vague. What is actually known from literature? Is it a direct effect on their transcription by binding to their promoters (e.g. via TEAD), or indirect through other signaling intermediates? Since nuclear export of the optogenetic YAP occurs on a 5 min timescale after illumination (Fig. 1) but Oct4 transcription doesn’t occur until 45 min after illumination (as assessed with MS2, Fig. 4), the regulation may not be direct.

On a related note, when quantifying the effect of YAP levels on Oct4/Nanog expression, the investigators report nuclear levels of YAP as the x-axis. There is certainly a strong sigmoidal relationship (Fig. 2C). However, YAP has functions in both the nucleus and the cytosol, so is it certain that the effect of YAP on Oct4/Nanog is mediated by nuclear YAP? Are nuclear and cytosolic YAP correlated to one another?

In addition, whether different levels of light (or YAP) are used, or flipping light on/off at certain frequencies, are used, different levels of Oct4 would presumably be achieved. So this is a simpler matter

of different Oct4 levels (or different levels of Oct4 concentration integrated over time) driving different cell fates, yes? Again, could be missing something.

While their optogenetic study is well-controlled, and low dose blue light is typically fine, it would be good to confirm that the light doses used in this study do not adversely affect the cells (i.e. affect cell proliferation or differentiation). Given this group's strong experience with optogenetics, this is likely not a concern.

The manuscript states that "mesodermal lineage induction requires mutually exclusive control (low Nanog and high Oct4, achievable at intermediate YAP or dynamic YAP inputs) or overlapping low expression (low Nanog and low Oct4, achievable at high YAP)." There should be references.

Reviewer #2 (Remarks to the Author):

The question and context of this paper are excellent - the goal is to arrive at a better understanding of how YAP transmits data into the nucleus. Do steady state levels control everything? Alternatively, are edges needed, are pulses needed, or, are periodic trains of pulses (oscillations) needed? Unfortunately the paper does not provide clear answers and it is uncertain if something has been discovered. The paper is thin on unambiguous results and very heavy on bold claims such as "Our data show optimal activation of YAP targets at a particular frequency of YAP activation" and "[we] provide a new operational framework for understanding how YAP controls specific cellular decisions." I do not recommend publication.

1/ According to the text, "23%- 50% of mESCs differentiating into the ecto- and mesodermal lineages exhibit discrete YAP pulses."

This is potentially interesting but hard to interpret. The figure shows six wavy lines labeled "sustained" and six wavy lines labeled "dynamic". There are statistical tests but they have not been done in a compelling way. If you take 10000 cells and measure anything, you will get wavy lines (microscope drift, cell movement, cell division, focus drift, changes of cell size during differentiation, changes of nuclear size during differentiation, changes of cell shape during differentiation, changes of cell aspect ratio, mitosis...).

The central data analysis question is, how do you take N wavy lines and convince yourself that you are dealing with a regular oscillation? Binarizing the traces and then running tests on the binarized data

would appear to leave you open to dozens of biases and artifacts - for example, how sensitive is the automated peak detector to different cell sizes/shapes, which are changing during differentiation? A fundamental claim of the paper, "YAP oscillates on timescales of 2.3-3h" appears to be based on the (unknown/uncertain) performance of an automated peak detection algorithm applied to noisy data from cells that are differentiating into different lineages? After the complex mean subtraction and data exclusion process that culminates with running the python find_peaks function with a setting of peak prominence=0.25 and height>=1.3, how sure are you that the data cannot be explained in any other way? How were those values chosen? How sensitive is the approach to different parameter settings? Given effective SNR of A, in what fraction of trials would you erroneously conclude that levels are oscillating despite the levels not actually oscillating? Where are the "discrete" YAP pulses in Fig. 1D?

2/ [External control of oscillation] Given the above, the main experimental purchase point of the paper - lets externally _impose_ an oscillation, is outstanding. That's precisely the right thing to do. Conceptually/hypothetically, a dream result would be that YAP-based cell differentiation is frequency controlled, with different outcomes requiring different oscillation frequencies, or something like that. Figure 6D asserts that shorter pulses (15-60 min) result in more cell proliferation compared to either no pulses ("chronic") or long pulses (>120 mins). A trivial alternative explanation for the data is that cells prefer "a little bit of light" at the right time. Also, the error bars of the 15 min pulse experiment overlap with the mean of the chronic pulse condition. Are the cells responding to a single edge? Is there even a significant effect of pulse vs. chronic? How would the cells respond to a total integrated light dose corresponding to the 15 min pulse schedule? I.e. could you achieve the same effect with intermediate light levels? The series of bold claims made in the entirety of Fig. 7 require equally dramatic and solid experimental data.

3/ Fig 7 - "Differential control of pluripotency factors and cellular decision-making through YAP levels and dynamics" seems almost entirely speculative given the above?

What I would recommend is that, whatever happens with this paper, you pick two foundational claims - such as "YAP oscillates on timescales of 2.3-3h" and "cell proliferation is pulse width modulated" and show through compelling experiments that those claims are in fact likely to be true. Once you have a solid foundation to stand on, further exploration is then warranted.

Reviewer #3 (Remarks to the Author):

This manuscript examines how different modes of YAP regulation can yield distinct outcomes in determining cell fate in mouse embryonic stem cells. YAP is known to regulate the choice between proliferation and differentiation in stem cells, however the exact mechanisms remain uncertain. YAP

was previously observed to periodically shift between a predominantly nuclear localization to the cytoplasm and back. These “pulses” were shown to be important for initiation of transcription in a breast epithelial cell line, but their function if any in a developmental context was not known. Here the authors show that mouse embryonic stem cells also show “pulses” of YAP nuclear export and import during differentiation. They use 2 approaches to precisely control the timing and magnitude of YAP activity by 1) using a tet-regulated promoter to control YAP levels, and 2) by using a “LEXY-YAP” fusion that can be exported from the nucleus in a light dependent manner. By manipulating overall YAP levels and/or the length and frequency of the “pulses” of nuclear YAP the authors were able to observe how distinct modes of YAP regulation could lead to different effects on differentiation and proliferation. For example different thresholds of YAP levels were required for repression of the differentiation markers Oct4 and Nanog. They also observed that certain pulse frequencies of nuclear YAP (similar to that observed with endogenous YAP) could induce Oct4 expression and proliferation, whereas differentiation requires persistently low level of YAP.

This work is a nice extension earlier work describing pulses of nuclear YAP, by showing that the pulses can trigger distinct developmental fates. The authors have created a clever system that allows them to delve into how the intricacies of YAP regulation can be decoded to dictate developmental fates. Although the magnitudes of many of the effects are small, the experiments are well controlled and reproducible. Although the manuscript does not address how YAP is regulated in a developmental context, it does point the way by showing which sorts of perturbations of YAP can yield distinct developmental outcomes. I have a few minor questions/suggestions.

1) The fold changes in Oct4 induced by YAP pulses in figure 3C is small. To assess the relevance of this fold change, it would be good to know how it compares to the normal change in Oct4 levels during differentiation in wild-type cells.

2) On page there are a couple conclusions that could use more explanation because it was not obvious how the authors reached the stated conclusion. The two examples are: “This suggests that YAP controls Oct4 expression through more than one regulatory entry point.”, and the sentence at end of the 3rd paragraph that starts with “Together, the results demonstrate that the adaptive change sensor and dose response module.....?”

3) It is not clear to me what is going on in Figure 5C-D. These panels are not referred to in the text.

Point-by-point response to the Reviewers

Reviewer #1:

1. The choice of time points used in the study sometimes has an unclear rationale. SNAP-YAP was analyzed 1.5 days after differentiation as a “a generally permissive time window for differentiation cues in mESCs.” This is a single time point, and it’s not clear whether cells at this time point are in the act of integrating YAP signal to influence the differentiation decisions later investigated in this study, or whether this is an arbitrary time point where the YAP dynamics are not relevant to the biology later studied. They then investigated the effect of their inducible YAP system (which is a clever system) on Oct4/Nanog expression at 2 days. Why use a different time point? Why use a single time point, when differentiation is a dynamic process?

Sorry for not providing a clear rationale for our choice of the differentiation time point for our experiments in the initial submission. The occurrence of endogenous YAP dynamics at 1.5d post differentiation motivated our functional dissection of YAP levels and dynamics around that time window. We assumed that the dynamics occur throughout a broader (12-24h) time window, similar to biochemical differentiation cues that typically act on time scales of days. To expand our investigation beyond this time window, we analyzed YAP dynamics at additional time points during early mesoderm differentiation (0.25-3d). Our results demonstrate that mESCs slowly acquire pulsatile YAP dynamics within 1.5d post differentiation induction (**Fig. S1E**). The dynamics are maintained from 1.5d post differentiation onwards until at least until 3d post differentiation. The pulse features (amplitude, duration) are comparable across that time window (1.5-3d post differentiation; **Fig. S1F,G**). The occurrence of YAP pulses during this broad time window justifies our choice of 1.5-2d post differentiation for a functional dissection of YAP dynamics. In addition, the time-resolved analysis of YAP dynamics should help enable replication of our differentiation conditions by others.

Figure S1E-G Quantification of endogenous YAP dynamics over the time course of early mesoderm differentiation. Shown are the percentage of cells exhibiting YAP pulses (E) and the amplitude (F) and duration (G) of those pulses at 1.5d and 3d post differentiation. Data for the naive condition and 1.5d post differentiation start in panels E-G are replicated from Fig. 1E, F and panel D. Shown are mean \pm SEM (E) and the Box and Whiskers with median and 5-95 Percentile (E,G), N=3 (E-G), p values from unpaired Student's t test (E).

2. The manuscript states “In response to an acute chronic input, adaptive signaling circuits transiently respond to the change in signal input but then reset to the initial baseline under sustained activation of the input.” “Acute chronic” is seemingly a contradictory phrase. In addition, the left panel of 3E shows a transient rather than a sustained activation of the input. This leads to some more general confusion about the proposed behavior of the system. Fig. 2C proposes to show that steady state nuclear levels of YAP drive different steady state levels of Oct4/Nanog. However, from the supplementary info, the experiment appears to involve spontaneously differentiating cells for 24 hours followed by illumination for 12 hours and measurement of Oct4/Nanog proteins at this 36 hour time point. Furthermore, Fig. 4 shows that a step increase in illumination (i.e. switching on the light) results in a pulse of Oct4 expression. In other words, constant illumination in Fig. 4 results in pulsatile Oct4 expression, all in the context of a cell that is undergoing spontaneous differentiation. How exactly is Fig. 2 then at steady state? Are the different Oct4 levels in Fig. 2C the result of the integral of different Oct4 transcriptional pulse sizes that happened 12 hours before, subtracting the effects of Oct4 protein and mRNA degradation during this 12 hours? In this case, how could Oct4 levels be referred to as steady state?

This brings up a significant concern with the following, a central conclusion of the manuscript: “Our results reveal two different YAP decoding modes by Oct4 that are implemented at the gene regulatory level: steady-state YAP levels control Oct4 in a dose-dependent manner, while acute YAP changes induce an adaptive transient response.” If a step increase to a new steady state of YAP led to a pulse of Oct4 transcription that was read out 12 hours later as a new Oct4 “steady state” (when in fact Oct4 protein and mRNA could be undergoing degradation to return back to the same original level, and thus Fig. 2c is a snap shot in time rather than a steady state), then the first half of the manuscript’s sentence could be a downstream, misinterpreted consequence of the second half of the sentence. Or is this reviewer missing something?

At the transcriptional level, a different question seemingly emerges. Different “steady state” (12 hour?) levels of YAP (Fig. 5C) were initially established as step changes in nuclear YAP. However, Oct4 transcriptional burst frequency transiently increases with a step change in YAP before returning to baseline in ~3 hours (Fig. 5D). So unless a step increase in YAP leads to an initial pulse and decay to baseline, followed in the next 9 hours (where the manuscript does not have experimental observations) in some increase to a higher steady state, these two outcomes are contradictory. That is, does a step change in YAP result in a long term increase in burst frequency (Fig. 5C) or a transient increase in burst frequency (Fig. 5D).

The central point of the Reviewer’s concern is that our experimental conditions do not enable us to distinguish between the importance of YAP levels versus YAP dynamics. Sorry for not making it more clear that we use two separate approaches for manipulating YAP dynamics (optogenetics) and steady-state levels (dox-inducible expression of YAP). The steady-state is achieved using a doxycycline inducible expression system to drive constant YAP levels over time scales of days. We make use of the expression heterogeneity between cells to map downstream responses (Oct4, Nanog) to different steady-state YAP levels. In contrast, YAP dynamics are induced through light illumination of our optogenetic YAP tool to acutely modulate YAP levels over time scales of minutes to hours (referred to as acute/dynamic YAP modulation). The optogenetic tool is also under doxycycline control to ensure comparability to our steady-state measurements. Generally, the doxycycline expression system induces maximum expression of YAP within approximately 12h. Our experimental setup (cell transfection and doxycycline addition, cell sorting, seeding and differentiation induction) takes a total of 3 days until we measure the effects of YAP expression

levels on Oct4 and Nanog expression or cell fate and proliferation at 1.5d post pluripotency exit. By 3d post doxycycline induction, we can assume that YAP levels are in steady-state. While the results in Figure 2 are based on the steady-state expression system, Figure 3 is based on the acute optogenetic modulation of YAP levels. We rewrote the paragraph introducing the inducible and optogenetic system to clearly distinguish the two systems.

Regarding the specific questions about the wording “acute chronic” and our schematic representation of acute chronic YAP export (Fig. 3E), we understand that this can be confusing. We were referring to the continuous YAP export with our optogenetic tool. The continuous export is driven by the acute onset of chronic light illumination, therefore referred to as “acute chronic”. To improve clarity, we added labels and headings for the schematic showing this (**Fig. 3E**).

Figure 3E) Possible decoding logic of dynamic YAP inputs through adaptation. For an adaptive system, continuous YAP export would transiently activate YAP effectors (left panel). In contrast, pulsed YAP inputs would induce sequential adaptive YAP responses; this would result in higher total Oct4 induction (center panel) than is seen for chronic YAP export (left panel). The adaptive system gives a maximum output at a specific pulse frequency input (center) compared to faster (right panel) or slower (not shown) frequencies.

Regarding the question about Oct4 transcription upon acute YAP export, we apologize that the callouts for Fig. 5C and D were missing in the main text. From the transcriptional analysis of our MS2 live imaging reporter, we find that an acute YAP export is accompanied by a transient increase in transcription burst frequency (**Fig. 5D**) that resets to the initial transcriptional state at ~ 3h post YAP export. Thus, despite cells having lower YAP levels, the transcriptional state (burst frequency) is comparable to the initial state under higher YAP levels. This is the feature of an adaptive system that acts as a change sensor and differs from the steady-state system in which YAP levels negatively correlate with burst frequency (**Fig. 5C**).

3. The manuscript states that “adaptive systems generate optimal responses at a specific input frequency that matches the resetting time.” It’s not clear what the word optimal means in this context. What is it optimal for?

We replaced the word “optimal response” with “maximal response” in the main text. We are referring to the response of a change sensor to dynamic inputs. Depending on the specific response time scale of an adaptive system, i.e. the initiation, duration and reset time of the response, a change sensor will only show a maximum integrated response when the timescale of the input matches the timescale of the resetting (an input that is too sustained gives only one pulse of output, a pulsatile input with infrequent pulses also gives a low response, but an intermediate oscillatory input gives multiple output pulses with a maximum integrated response).

4. Can the authors describe molecularly how YAP is regulating Oct4/Nanog? This question isn't even broached until the end of page 6 of the manuscript, and the wording is somewhat vague. What is actually known from literature? Is it a direct effect on their transcription by binding to their promoters (e.g. via TEAD), or indirect through other signaling intermediates? Since nuclear export of the optogenetic YAP occurs on a 5 min timescale after illumination (Fig. 1) but Oct4 transcription doesn't occur until 45 min after illumination (as assessed with MS2, Fig. 4), the regulation may not be direct.

We agree with the reviewer that the delayed onset of the adaptive response to acute YAP export might indicate an indirect mechanism of YAP-dependent Oct4 regulation. Based on the literature and genome annotation, evidence for both direct and indirect mechanisms of YAP-dependent Oct4 regulation exists. The Oct4 regulatory region harbors four different putative TEAD/YAP binding sites that suggest a direct regulation. Previous reports have identified direct YAP/TEAD interaction with the Oct4 locus through ChIPseq (Lian et al. 2010; Beyer et al. 2013; Tamm, Böwer, and Annerén 2011). On the other hand, YAP also controls regulators of Oct4, most importantly Sox2 and Nanog, as well as global chromatin modifications that could indirectly affect Oct4 expression (Passaro, JCB, 2021). We now briefly mention the direct regulation in the main text when we introduce YAP-dependent Oct4/Nanog regulation. To address this question experimentally, we used a previously-developed Oct4 reporter that harbors the entire Oct4 regulatory region (distal enhancer, proximal enhancer, promoter) and in which the Oct4 coding region is replaced by GFP (Gafni et al. 2013). We deleted all four putative TEAD binding sites and repeated our steady-state dose-response analysis of Fig. 2C. The results demonstrate that the absence of the TEAD sites does not interfere with the repressive activity of YAP. However, it alters the shape of the sigmoidal repression curve and renders the locus more sensitive to lower YAP levels. Comparison of the Hill coefficients suggests that the TEAD sites establish the negative cooperativity of the threshold response. This demonstrates that under steady-state conditions, the threshold behavior of Oct4 repression is likely mediated through direct interaction of YAP/TEAD with the Oct4 locus. We added the results as **Fig. 2D** and in the main text. We thank the reviewer for this suggestion, which should be of interest to the YAP field.

Figure 2D) Deletion of the TEAD binding sites in the gene regulatory region of the Oct4 locus reporter affect the sensitivity of the locus to steady-state YAP levels, shifting the Hill coefficient from 1.6 ± 0.2 (WT) to 0.9 ± 0.1 (TEAD mutant). The Oct4 reporter is ectopically introduced and based on the replacement of the Oct4 open reading frame with GFP (panel D, top). The location of the TEAD binding sites is schematically shown on top, see Fig S5A for details. Shown are Oct4 levels as a function of nuclear YAP levels and the Hill curve fit from a single experiment. See Fig. 5C for replicates. Hill coefficient represents mean \pm SEM from $N=3$. p values from unpaired Student's t test.

5. On a related note, when quantifying the effect of YAP levels on Oct4/Nanog expression, the investigators report nuclear levels of YAP as the x-axis. There is certainly a strong sigmoidal relationship (Fig. 2C). However, YAP has functions in both the nucleus and the cytosol, so is it certain that the effect of YAP on Oct4/Nanog is mediated by nuclear YAP? Are nuclear and cytosolic YAP correlated to one another?

Both our optogenetic and doxycycline induction system show a strong correlation of nuclear and cytoplasmic YAP levels and can not distinguish between a primary functional role in the nucleus versus cytosol. Light-gated YAP export proportionally decreases nuclear YAP levels and increases the cytoplasmic pool. Similarly, doxycycline-induced expression of YAP or LEXY-YAP increases levels in both compartments. To address the question, we instead made use of the endogenous system which establishes a heterogeneous population of nuclear and cytoplasmic YAP levels during spontaneous differentiation (Fig. S4C). We quantify nuclear and cytoplasmic YAP levels and quantitatively compare the effect of high nuclear/low cytoplasmic and low nuclear/high cytoplasmic cells on Oct4 protein levels. We find that YAP's repressive effect on Oct4 levels correlates much more strongly with nuclear YAP than cytoplasmic YAP (Fig. S4 D,E). While we cannot exclude that the cytoplasmic YAP pool affects Oct4/Nanog levels, our correlation strongly suggest that the nuclear YAP pool is the predominant regulator of the sigmoidal relationship observed in Fig. 2C. We added the result to the text.

Figure S4C-E) Role of cytoplasmic vs. nuclear YAP for Oct4 repression by comparison of Oct4 levels in differentiating mESCs with high nuclear and low cytoplasmic YAP (panel C, solid yellow line; panel D, grey bars), or low nuclear and high cytoplasmic YAP (panel C, dashed yellow line; panel D, red bars). The YAP and Oct4 signal were quantified from IF stainings as shown in (C), and cells were selected for comparable YAP levels in the opposing compartment (D). Oct4 levels are significantly lower in presence of high nuclear YAP than comparable cytoplasmic levels (E). Shown are mean +/- SEM, N=3 (D,E). p values from unpaired Student's t test (E).

6. In addition, whether different levels of light (or YAP) are used, or flipping light on/off at certain frequencies, are used, different levels of Oct4 would presumably be achieved. So this is a simpler matter of different Oct4 levels (or different levels of Oct4 concentration integrated over time) driving different cell fates, yes? Again, could be missing something.

How steady-state and dynamic decoding capacities of Oct4 differ is an important point that was insufficiently discussed in our primary submission. We recognize that our observation that steady-state YAP expression demonstrates a clear responsiveness of Oct4 to YAP, but our light-gated chronic YAP export for 12h did not show a response may have been confusing. We therefore performed a more detailed analysis of the Oct4 response to our chronic light inputs. By comparing the shape of the Hill curve pre and post illumination, we find that chronic YAP export induces a minor shift of the Hill curve, but this only occurs at lower YAP levels than observed for oscillatory inputs (**Fig. S6A**). This was missed in our previous analysis that only included cells that were at least moderately repressed in the dark state (cutoff at IC50 of the dark control). We now added this data that shows that continuous YAP export can derepress Oct4 (**Fig. S6B**), but oscillatory inputs are significantly more potent at inducing Oct4 at higher LEXY-YAP levels (**Fig. 3C**). In response to the reviewer's question, we also probed if time-integrated readouts can explain the observed Oct4 response behavior to pulsatile YAP inputs. We compare a number of light patterns that use the same total light dose as the dynamic input that induces Oct4 (240min ON, 60min OFF), as well as chronic light inputs with an integrated light dose comparable to the pulsatile condition (**Fig. S7B**). To titrate the chronic light dose, we established the YAP dose-response curve by IF staining (**Fig. S7A**). We find that the oscillatory inputs significantly differ in their potency in Oct4 induction despite using the same integrated light, demonstrating that some aspect of the dynamic input beyond simple integration over time is necessary to explain the response to dynamical inputs. Similarly, none of the lower chronic light conditions could achieve Oct4 induction comparable to the oscillatory input. Importantly, the Oct4 quantification includes cells with lower YAP levels (cutoff at IC5 of the dark control) to ensure we are not missing chronic responses at lower levels. Together, our results demonstrate that the Oct4 signaling module has two different

decoding capacities for steady-state and dynamic inputs. The dynamic response differs from a time averaged or integrated decoder. The results are reminiscent of the response behavior for YAP-dependent proliferation (Fig. 6D) and indicate that a dual decoding strategy of steady-state levels versus dynamics may be a general feature of YAP signaling. A response to dynamics as well as steady-state levels could enable cells to modulate YAP effectors at a broader range of timescales and levels of YAP activation than would be possible with either strategy alone.

Figure S6 Quantification of Oct4 induction upon light gated control of YAP export

A) Hill curve representation of the light-gated Oct4 induction shown in Fig. 3C. Shown are Oct4 protein levels as a function of nuclear LEXY-YAP/YAP levels upon chronic (top left) and pulsed (240min On/60min OFF; top, right) illumination in LEXY-YAP (top) and YAP control cells (bottom). All light conditions (blue) are shown in reference to a dark control (grey). Shown are mean from N=7 and the Hill curve fits with 95% CI. Dashed lines indicate the IC50 of the dark control condition. Red arrows indicate Oct4 induction upon pulsed illumination (broad range of YAP levels) or chronic light (small range of YAP levels). **B)** Quantification of Oct4 protein levels upon chronic or pulsed (240 min ON/60min OFF) light as shown in Fig. 2C but including cells with lower YAP levels expression levels. Shown are mean +/- SEM. p values from unpaired Student's t test.

Figure S7 Probing the dynamic decoding capacity of Oct4 by light pulse and dose modulation.

A) Light dose titration of our LEXY-YAP tool by quantification of YAP levels upon illumination with different light intensities. YAP was quantified from IF stainings. Shown are mean \pm SEM, $N=5$. 40% export refers to the light used throughout our work to achieve full export. Note that cell fixation slightly affects nuclear YAP levels (full light = 40% YAP export) as compared live cell measurements (Fig. 11, full light \sim 55% export). **B**) Light profiles used to test the dynamic decoding capacity of Oct4. Shown are illumination profiles for pulse modulation (top row) and light dose modulation. Pulse modulation conditions use the same total light dose (right y-axis, blue line) but differ in their pulse durations. Light dose modulation conditions use the same light pattern (chronic light) but differ in the amount of light. 33% YAP export relates to the same total amount of light as the oscillatory 240 min ON/ 60mn OFF light pattern. Light intensities refer to the YAP export values indicated by vertical dashed lines in (A). **C**) Quantification of Oct4 protein induction upon illumination with light pulse and dose modulation patterns shown in (B). Results demonstrate that Oct4 induction by our oscillatory YAP pattern (240 min ON, 60min OFF) cannot be explained by decoding of the integral light intensity. Shown are mean \pm SEM, $N=5$. p values from unpaired Student's t test (C) or paired t-test (indicated).

7. While their optogenetic study is well-controlled, and low dose blue light is typically fine, it would be good to confirm that the light doses used in this study do not adversely affect the cells (i.e. affect cell proliferation or differentiation). Given this group's strong experience with optogenetics, this is likely not a concern.

We have put a lot of effort into the design of our optogenetic tool and light illumination devices to ensure that blue light illumination is not inducing adverse effects on cell physiology. This includes the use of a previously reported LOV2 mutant (LEXYV416I, (Kögler et al. 2021)) that is more sensitive to blue light than the WT protein, illumination with pulsed light inputs (1 sec on, 2 sec off) to reduce the total light exposure, the use of a previously established image denoising pipeline to image with very low laser intensities for our live-imaging experiments, and a fan-driven LED illumination device that reduces thermal toxicity of LED illumination for differentiation and proliferation assays. To control for phototoxicity, all our optogenetic experiments (**Fig.3 C,D**, **Fig. 4D**, **Fig.6C,D**) include a non-light responsive control (SNAP-YAP) that is cultured and illuminated under the same light conditions as our optogenetic cells. For all readouts (Oct4 protein levels, Oct4 transcription, proliferation, differentiation), we demonstrate negligible effects of the light illumination compared to the dark control.

8. The manuscript states that "mesodermal lineage induction requires mutually exclusive control (low Nanog and high Oct4, achievable at intermediate YAP or dynamic YAP inputs) or overlapping low expression (low Nanog and low Oct4, achievable at high YAP)." There should be references.

Thank you for pointing this out. We have added references.

Reviewer #2:

The question and context of this paper are excellent - the goal is to arrive at a better understanding of how YAP transmits data into the nucleus. Do steady state levels control everything? Alternatively, are edges needed, are pulses needed, or, are periodic trains of pulses (oscillations) needed? Unfortunately the paper does not provide clear answers and it is uncertain if something has been discovered. The paper is thin on unambiguous results and very heavy on bold claims such as "Our data show optimal activation of YAP targets at a particular frequency of YAP activation" and "[we] provide a new operational framework for understanding how YAP controls specific cellular decisions." I do not recommend publication.

9. According to the text, "23%- 50% of mESCs differentiating into the ecto- and mesodermal lineages exhibit discrete YAP pulses." This is potentially interesting but hard to interpret. The figure shows six wavy lines labeled "sustained" and six wavy lines labeled "dynamic". There are statistical tests but they have not been done in a compelling way. If you take 10000 cells and measure anything, you will get wavy lines (microscope drift, cell movement, cell division, focus drift, changes of cell size during differentiation, changes of nuclear size during differentiation, changes of cell shape during differentiation, changes of cell aspect ratio, mitosis...).

The central data analysis question is, how do you take N wavy lines and convince yourself that you are dealing with a regular oscillation? Binarizing the traces and then running tests on the binarized data would appear to leave you open to dozens of biases and artifacts - for example,

how sensitive is the automated peak detector to different cell sizes/shapes, which are changing during differentiation? A fundamental claim of the paper, “YAP oscillates on timescales of 2.3-3h” appears to be based on the (unknown/uncertain) performance of an automated peak detection algorithm applied to noisy data from cells that are differentiating into different lineages? After the complex mean subtraction and data exclusion process that culminates with running the python find_peaks function with a setting of peak prominence=0.25 and height>=1.3, how sure are you that the data cannot be explained in any other way? How were those values chosen? How sensitive is the approach to different parameter settings? Given effective SNR of A, in what fraction of trials would you erroneously conclude that levels are oscillating despite the levels not actually oscillating? Where are the “discrete” YAP pulses in Fig. 1D?

We agree that our paper would benefit by a demonstration of the robustness of our peak finding algorithm and explanation of our parameter choice. In the revision, we explored the performance of our analysis pipeline by comparing endogenous YAP dynamics to two different negative controls. Our first control is chemically fixed endogenous YAP reporter cells with the same signal-to-noise as our live imaging conditions to test for potential effects of focus drift, bleaching or image denoising. Our second control is our endogenous YAP reporter line stably co-expressing GFP at low levels. This allows us to directly compare YAP to a non-dynamic fluorophore by simultaneous live imaging under the same differentiation and imaging conditions as used for our experiments in **Fig. 1**. The simultaneous imaging approach controls for potential effects of cell movement, changes of cell size, nuclear size, cell shape and mitosis during differentiation, as mentioned by the reviewer. We expressed GFP at very low levels and imaged it at comparable single-to noise levels to enable comparison with our YAP experiments (**Fig. S2 B**). By mapping the percentage of cells exhibiting YAP or GFP pulses (**Fig. S2E,F**), we demonstrate that our parameter choice yields a false positive rate of ~4% (2.9% of GFP traces are detected as pulsing vs 73% of YAP cells are detected as pulsing; **Fig. S2 F**). This is within the error of our measurement and verifies that the observed YAP pulses are not a technical or experimental artifact. The heatmap in **Fig S2 D** further shows that our peak detection algorithm detects YAP pulses over a much wider range of parameter settings than the GFP negative control, demonstrating the robustness of our approach.

Figure S2 Performance verification of the peak detection algorithm

A) The peak detection algorithm identifies local maxima (peak prominence, red arrow) by comparison of neighboring values and further filters identified peaks by the absolute peak height (blue arrow). **B)** Quantitative comparison of the absolute GFP intensity to the endogenous YAP signal demonstrates comparable signal-to-noise for both reporters. **C-H)** Verification of our peak detection approach by comparison of endogenous YAP pulses to fixed cells expressing endogenous SNAP-YAP (panel C, E, G) and simultaneous live-imaging of a non-dynamic fluorophore (GFP) and endogenous YAP (panel D, F, H) at comparable signal-to-noise (see B). Heatmaps (C, D) show the percentage of cells pulsing as a function of peak prominence and peak height. Shown are mean, $N=3$. Panels E and F refer to the white dashed line in panels C, D and show the effect of peak prominence for pulse detection at fixed peak height = 1.3. Our parameter choice (prominence = 0.25, peak height = 1.3, red shading in panels E,F) yields a false positive rate of 4% (GFP control: 2.9% of cells pulsing; YAP: 73% of cells pulsing). Example traces of all conditions (fixed YAP, GFP control and endogenous YAP) are shown in G, H. The GFP and YAP traces were acquired simultaneously in the same cells (see cell ID label). Peak height and width are indicated by horizontal and vertical red lines. Panel E,F show mean \pm SEM, $N=3$.

10. *[External control of oscillation] Given the above, the main experimental purchase point of the paper - lets externally _impose_ an oscillation, is outstanding. That's precisely the right thing to do. Conceptually/hypothetically, a dream result would be that YAP-based cell differentiation is frequency controlled, with different outcomes requiring different oscillation frequencies, or something like that. Figure 6D asserts that shorter pulses (15-60 min) result in more cell proliferation compared to either no pulses ("chronic") or long pulses (>120 mins). A trivial alternative explanation for the data is that cells prefer "a little bit of light" at the right time. Also, the error bars of the 15 min pulse experiment overlap with the mean of the chronic pulse condition. Are the cells responding to a single edge? Is there even a significant effect of pulse vs. chronic? How would the cells respond to a total integrated light dose corresponding to the 15 min pulse schedule? I.e. could you achieve the same effect with intermediate light levels? The series of bold claims made in the entirety of Fig. 7 require equally dramatic and solid experimental data.*

This important question is related to the comment by Reviewer #1, point 6 (please also see the answer above). Given the time constraints of the revision, we could not dissect all YAP-dependent responses and decided to focus on YAP modulation of Oct4 to understand the role of dynamics in YAP effector regulation. For this purpose, we used pulse modulation and light dose titration (**Fig. S7B**) to probe if time integrated readouts can explain the observed Oct4 phenotype to oscillatory YAP inputs (240 min ON/60 min OFF pulse, **Fig. 3C**). We demonstrate that 60 min YAP pulses with a reset time of 240 min induced significantly more Oct4 protein levels than conditions that use the same total light input but at different dynamics (**Fig. S7C**). In line with this, comparable chronic export using the same time integrated light inputs as the oscillatory inputs cannot sufficiently induce Oct4. This demonstrates that the Oct4 decoding module cares about specific dynamic features and not about a little bit of light or a single edge. Important to note here is that we performed a detailed analysis of the Oct4 response to chronic YAP input at full light dose to understand the seemingly contradictory result that Oct4 correlates with steady-state YAP levels but was unaffected by continuous 12h YAP export. A more detailed analysis of the chronic YAP export condition showed that chronic light indeed induces a minor Oct4 response (**Fig. S6A**), yet only at lower YAP levels compared to the oscillatory inputs (see response to Reviewer #1, point 6 for more details). We initially missed this point since we quantified Oct4 induction at higher YAP levels where phenotypes from oscillatory YAP dynamics dominate. Our pulse and light dose modulation results now include the lower LEXY-YAP expression levels to ensure that are not missing responses from lower chronic inputs in that range. Together our results show that the Oct4 module has dual decoding capacity of steady-state and dynamic YAP inputs. We edited the text to clearly distinguish these two modes of YAP decoding.

Regarding YAP-dependent proliferation: The main point of Figure 6 was to compare the differentiation (**Fig. 6C**) vs. proliferation (**Fig. 6D**) response to YAP inputs. As stated in the main text, "[...] the proliferative response reveals a complementary decoding logic to that of cellular differentiation". While both respond to chronic light inputs, differentiation rejects YAP dynamics, whereas proliferation is responsive to oscillatory YAP inputs. This establishes a signaling mode (dynamics) that can exclusively talk to the proliferation module. We have made this point clearer in this paragraph in the revised manuscript.

A

B

Figure S6 Quantification of Oct4 induction upon light gated control of YAP export

A) Hill curve representation of the light-gated Oct4 induction shown in Fig. 3C. Shown are Oct4 protein levels as a function of nuclear LEXY-YAP/YAP levels upon chronic (top left) and pulsed (240min On/60min OFF; top, right) illumination in LEXY-YAP (top) and YAP control cells (bottom). All light conditions (blue) are shown in reference to a dark control (grey). Shown are mean from N=7 and the Hill curve fits with 95% CI. Dashed lines indicate the IC50 of the dark control condition. Red arrows indicate Oct4 induction upon pulsed illumination (broad range of YAP levels) or chronic light (small range of YAP levels). **B**) Quantification of Oct4 protein levels upon chronic or pulsed (240 min ON/60min OFF) light as shown in Fig. 2C but including cells with lower YAP levels expression levels. Shown are mean +/- SEM. p values from unpaired Student's t test.

Figure S7 Probing the dynamic decoding capacity of Oct4 by light pulse and dose modulation.

A) Light dose titration of our LEXY-YAP tool by quantification of YAP levels upon illumination with different light intensities. YAP was quantified from IF stainings. Shown are mean \pm SEM, N=5. 40% export refers to the light used throughout our work to achieve full export. Note that cell fixation slightly affects nuclear YAP levels (full light = 40% YAP export) as compared live cell measurements (Fig. 11, full light \sim 55% export). **B)** Light profiles used to test the dynamic decoding capacity of Oct4. Shown are illumination profiles for pulse modulation (top row) and light dose modulation. Pulse modulation conditions use the same total light dose (right y-axis, blue line) but differ in their pulse durations. Light dose modulation conditions use the same light pattern (chronic light) but differ in the amount of light. 33% YAP export relates to the same total amount of light as the oscillatory 240 min ON/ 60mn OFF light pattern. Light intensities refer to the YAP export values indicated by vertical dashed lines in (A). **C)** Quantification of Oct4 protein induction upon illumination with light pulse and dose modulation patterns shown in (B). Results demonstrate that Oct4 induction by our oscillatory YAP pattern (240 min ON, 60min OFF) cannot be explained by decoding of the integral light intensity. Shown are mean \pm SEM, N=5. p values from unpaired Student's t test (C) or paired t-test (indicated).

11. Fig 7 - “Differential control of pluripotency factors and cellular decision-making through YAP levels and dynamics” seems almost entirely speculative given the above?

What I would recommend is that, whatever happens with this paper, you pick two foundational claims - such as “YAP oscillates on timescales of 2.3-3h” and “cell proliferation is pulse width modulated” and show through compelling experiments that those claims are in fact likely to be true. Once you have a solid foundation to stand on, further exploration is then warranted.

We extensively validated our peak detection algorithm (**Fig. S2**) to demonstrate that our results are not a technical artifact. In addition, we performed a time resolved analysis of YAP dynamics during pluripotency exit (**Fig. S1E**) that shows the progressive acquisition of YAP dynamics during differentiation. We hope this convinces the reviewer that the observed YAP dynamics are significant. Using pulse and light dose modulation (**Fig. S7**), we probe the dynamic decoding mode of Oct4 for a possible alternate time-integrated readout. We demonstrate that the total light dose cannot explain the Oct4 response to oscillatory YAP inputs. This is further supported by our transcriptional readouts, which show that the response of Oct4 gene expression to acute YAP export is adaptive (**Fig. 4D**). These observations provide a framework for understanding how consecutive YAP pulses can be decoded.

Reviewer #3:

12. *The fold changes in Oct4 induced by YAP pulses in figure 3C is small. To assess the relevance of this fold change, it would be good to know how it compares to the normal change in Oct4 levels during differentiation in wild-type cells.*

We followed the reviewer’s suggestion and compared the magnitude of our YAP-dependent Oct4 phenotypes to endogenous Oct4 levels during differentiation (up to 4d post pluripotency exit) by Oct4 IF staining. These data show that endogenous Oct4 levels drop by ~73 % within 4d of spontaneous differentiation in comparison to the naive condition. The regulatory range of our steady-state YAP measurements (switch-like repression, **Fig. 2C**) falls within this range of endogenous Oct4 levels (**Fig. S6C**). The depressed state (very low YAP expression) is comparable to Oct4 levels in the naïve/early differentiating state (83% of Oct4 levels in the naive state), while the repressed state (> IC75) reduces Oct4 protein below the levels present at four days post differentiation, in comparison to only ~4% of the Oct4 levels in the naive state.

We agree that a fold-change readout of Oct4 phenotypes in response to our optogenetic perturbations is difficult to interpret. We therefore changed the way we quantify Oct4 levels. We now normalize Oct4 levels by the derepressed state of the steady-state dark control, which we mapped to the endogenous physiological Oct4 range (see above). We then express Oct4 induction upon perturbation as difference within the normalized range (Δ Oct4, **Fig. S6C**). The magnitude of the Oct4 response to dynamic YAP inputs for 12h (Δ Oct4 =0.25, **Fig. 3C**) compares the amount of Oct4 protein lost during 1d of differentiation.

Figure S6C C) Relation of our YAP-dependent Oct4 phenotypes to the range of endogenously-observed Oct4 levels. Following pluripotency exit, mESCs decrease endogenous Oct4 levels by ~73% over a time course of 4 days post spontaneous differentiation. Our steady-state measurements of Oct4 repression (Fig. 2C) compares to ~83% (low YAP levels) and 4% (high YAP levels) of the Oct4 protein found in naive cells (indicated by horizontal dashed line, red shading represents the SEM of that measurement, N=3). The magnitude of our optogenetic Oct4 induction through oscillatory YAP dynamics (Fig. 2C, pulsed light, 240min ON/ 60min OFF, $\Delta\text{Oct4} = 0.25 \pm 0.1$) is comparable to the amount of Oct4 protein lost within 1d during spontaneous differentiation. Oct4 levels were quantified from IF stainings. Shown are mean \pm SEM.

13. On page there are a couple conclusions that could use more explanation because it was not obvious how the authors reached the stated conclusion. The two examples are: "This suggests that YAP controls Oct4 expression through more than one regulatory entry point.", and the sentence at end of the 3rd paragraph that starts with "Together, the results demonstrate that the adaptive change sensor and dose response module.....?"

We have modified the main text to better explain how we reached these conclusions.

14. It is not clear to me what is going on in Figure 5C-D. These panels are not referred to in the text.

Thanks for pointing this out. Callouts to Figure 5C-D were missing and have now been added to the main text.

References cited

- Beyer, Tobias A., Alexander Weiss, Yuliya Khomchuk, Kui Huang, Abiodun A. Ogunjimi, Xaralabos Varelas, and Jeffrey L. Wrana. 2013. "Switch Enhancers Interpret TGF- β and Hippo Signaling to Control Cell Fate in Human Embryonic Stem Cells." *Cell Reports* 5 (6): 1611–24.
- Gafni, Ohad, Leehee Weinberger, Abed Alfatah Mansour, Yair S. Manor, Elad Chomsky, Dalit Ben-Yosef, Yael Kalma, et al. 2013. "Derivation of Novel Human Ground State Naive Pluripotent Stem Cells." *Nature* 504 (7479): 282–86.
- Kögler, Anna C., Yacine Kherdjemil, Katharina Bender, Adam Rabinowitz, Raquel Marco-Ferreres, and Eileen E. M. Furlong. 2021. "Extremely Rapid and Reversible Optogenetic Perturbation of Nuclear Proteins in Living Embryos." *Developmental Cell* 56 (16): 2348–2363.e8.
- Lian, Ian, Joungmok Kim, Hideki Okazawa, Jiagang Zhao, Bin Zhao, Jindan Yu, Arul Chinnaiyan, et al. 2010. "The Role of YAP Transcription Coactivator in Regulating Stem Cell Self-Renewal and Differentiation." *Genes & Development* 24 (11): 1106–18.
- Tamm, Christoffer, Nathalie Böwer, and Cecilia Annerén. 2011. "Regulation of Mouse Embryonic Stem Cell Self-Renewal by a Yes-YAP-TEAD2 Signaling Pathway Downstream of LIF." *Journal of Cell Science* 124 (Pt 7): 1136–44.

REVIEWER COMMENTS

Reviewer #1 (Remarks to the Author):

The authors have conducted additional experiments and introduced text edits to address the concerns of this reviewer. In particular, the question of whether the effects of pulsatile YAP activity are simply mediated by cells integrating the area under the "pulse curve" and reading this level out as an effective steady state at a higher YAP level, i.e. do pulse dynamics matter, is much better addressed in the revised manuscript than in the original. The following summary in the rebuttal of new data was most effective in addressing this key point: "We find that the oscillatory inputs significantly differ in their

potency in Oct4 induction despite using the same integrated light, demonstrating that some aspect of the dynamic input beyond simple integration over time is necessary to explain the response to dynamical inputs."

Regarding this reviewer's comment about whether this system is truly at steady state, the comment was meant to address the fact that this is complex system that is continuously changing as cells progressively undergo differentiation while various YAP perturbations are being introduced. The authors should be a little careful in assuming that a fixed level of dox will induce a truly steady state / constant level of YAP over the course of the experiment, given that the cells are dynamically and dramatically changing their transcriptional state and activity throughout the experiment.

That said, the concerns of this reviewer have been largely addressed.

Reviewer #3 (Remarks to the Author):

The authors have fully addressed my concerns.

Reviewer #4 (Remarks to the Author):

This study quantifies changes in YAP dynamics during the differentiation of naïve mESCs to mesendoderm or ectoderm and perturbs system dynamics using elegant and technically impressive optogenetic control and image-based readouts. The authors report interesting changes in YAP dynamics

during differentiation (from sustained to oscillating) and show that different YAP target genes Oct4 and Nanog have different sensitivities to YAP levels (which also depend on the cellular state), that transcription at the Oct4 locus is transient in response to a decrease in nuclear YAP, and that YAP dynamics modulation during differentiation can affect cell fate proportions. This study attempts to explain how different YAP signaling dynamics can direct different target gene and cell fate outcomes, and the results are of broad interest to the fields of optogenetics, signaling dynamics, and developmental biology. However, a few concerns remain regarding the interpretation of results and the lack of experimental support for certain claims of the study.

This review is to specifically assess the authors' responses to Reviewer #2 criticism.

9. The authors added important controls for the quantification of oscillatory YAP dynamics, and particularly the addition of an internal GFP control strengthens their claims.

10. The added figure S7 provides important controls to support the authors' claim that Oct4 is more strongly induced by an oscillatory YAP rather than a continuous YAP signal. However, could the authors please clarify why the constant 40% YAP export condition in Fig S7C differs greatly from the "chronic" condition in Fig 3C? The data in Fig S7C seems to indicate that comparable Oct4 expression can be achieved both with oscillatory YAP dynamics and 40% "chronic"/continuous light illumination, which would undermine the key conclusion that "oscillatory dynamic YAP inputs more efficiently induce Oct4 and proliferation than do sustained inputs".

Another small comment is that the use of many different terms to describe similar (though sometimes importantly different) conditions is confusing e.g. "light dose modulation" (Fig S7), "sustained" (Fig 1), "concentrations" (Fig 7), "levels" (Fig 6), "steady-state" (Fig 5), "chronic" (Fig 3). These terms are sometimes used interchangeably (e.g. levels, steady-state, and concentrations). It would help if the authors define these terms clearly and use them consistently in the proper context.

11. The authors have added important controls to support their claims, but a few concerns still remain on the interpretation of results. In general, the manuscript would benefit from clarification and rephrasing to reduce overstatement and speculation. I urge the authors to carefully go through the claims they are making in the manuscript and correct the wording to match the experimental evidence.

Firstly, it is difficult to interpret the results of Figure 2 and Figure 6 without a clearer understanding of the experimental setup. It is stated that the mESCs are undergoing "spontaneous" and "undirected differentiation". How is this different from the mesoderm and ectoderm differentiation regimes of Fig 1? What are the cell lineages and cell type proportions under WT unperturbed conditions? If the cells are undergoing spontaneous differentiation, could it be that the majority of cells develop oscillatory YAP

signaling, as in Figure 1? Even though the inducible YAP system expresses YAP at constant dox-controlled gene expression levels, it is conceivable that oscillations (ex. in nuclear YAP amounts via nuc/cyt shuttling) can still emerge during differentiation due to the corresponding changes in cell state. Therefore, is static analysis (as opposed to live imaging) the correct modality for correlating YAP intensity with Oct4/Nanog levels in Figure 2? Further, can the different media conditions used in Figure 6 be affecting the cell's ability to "decode" the inducible/optogenetically controlled YAP dynamics? I recommend the authors to clarify this point in the text and explicitly state the assumptions they are making.

Secondly, the authors repeatedly refer to a cell "decoding" YAP levels, and that "individual YAP target genes differentially decode YAP levels and dynamics". However, this claim is worded inconsistently throughout the manuscript and seems to lack strong experimental evidence. After Fig 3 the authors state "Together, our results demonstrate that the Oct4 signaling module has two different decoding capacities for steady-state and dynamical inputs", whereas after Fig 5 the authors state "This suggests that observed differences in the interpretation of steady-state and dynamic YAP inputs are likely established through other regulatory feedback differences, possibly at the level of YAP itself." Which module, then, is responsible for this differential decoding (confusingly worded as 'differences in the interpretation') - is it Oct4 (due to YAP binding to its regulatory regions), YAP itself, or some other cell processes/regulatory feedback? It would greatly improve the clarity of the paper if the authors can clearly explain what they mean by "decoding" and "differential decoding" and to stay consistent about which process they hypothesize is performing the "decoding".

On a related topic, could the authors please clarify exactly which experimental evidence allows them to make the central claim that YAP signals are "differentially decoded" by Oct4? From the data presented, it rather supports a conclusion that YAP dynamics are differentially encoded depending on cellular state (proliferation vs. differentiation), and are subsequently consistently decoded via YAP at Oct4 regulatory regions. Indeed, YAP sensitivity is clearly dependent on cell state (Fig S4A vs. 2C), and YAP dynamics change depending on the cell state changes associated with differentiation (Fig 1), which suggests that YAP dynamics are encoded by cell state or environmental cues. It seems the YAP perturbation experiments in Fig 6 are performed during mESC "spontaneous" differentiation conditions, so the question remains: are YAP dynamics instructive to fate specification, or are they permissive and a consequence of (and responding to) fate specification? The broad claims of Fig 7 imply they are instructive cues, and further the authors claim that "cells can titrate YAP doses". However, can YAP dynamics instead be a response to the broad cellular state changes (signaling, metabolic, cell cycle, mechanics, epithelial-to-mesenchymal transition) or population context (cell density, cell contacts) that occur during mESC differentiation? Could the differentiation compounds (CHIR, retinoic acid, or FBS) induce differences in YAP dynamics and the observed switch from sustained to oscillatory, without a cell actively "titrating YAP doses"? I would like the authors to comment on these questions in the Discussion section and amend their strong claims in the text accordingly.

Point-by-point response to the Reviewers

We thank the Reviewers for their constructive feedback on our revision. Below we provide a detailed point-by-point response to their remaining questions and suggestions.

Reviewer #1 (Remarks to the Author):

The authors have conducted additional experiments and introduced text edits to address the concerns of this reviewer. In particular, the question of whether the effects of pulsatile YAP activity are simply mediated by cells integrating the area under the "pulse curve" and reading this level out as an effective steady state at a higher YAP level, i.e. do pulse dynamics matter, is much better addressed in the revised manuscript than in the original. The following summary in the rebuttal of new data was most effective in addressing this key point: "We find that the oscillatory inputs significantly differ in their potency in Oct4 induction despite using the same integrated light, demonstrating that some aspect of the dynamic input beyond simple integration over time is necessary to explain the response to dynamical inputs."

Thanks for the suggestion. This was an important point to substantiate.

Regarding this reviewer's comment about whether this system is truly at steady state, the comment was meant to address the fact that this is complex system that is continuously changing as cells progressively undergo differentiation while various YAP perturbations are being introduced. The authors should be a little careful in assuming that a fixed level of dox will induce a truly steady state / constant level of YAP over the course of the experiment, given that the cells are dynamically and dramatically changing their transcriptional state and activity throughout the experiment.

We cannot rule out that endogenous YAP dynamics contribute to the concentration-dependent effect we observe in **Fig. 2C**. However, the evident correlation (Oct4 $R^2=0.96$; Nanog $R^2 = 0.97$) between nuclear YAP levels and Oct4/Nanog protein levels (**Fig. 2C**) suggest a strong concentration-dependency. In addition, the naïve (2i+LIF) condition, in which we do not detect substantial YAP dynamics (**Fig. 1E**), shows a similar shape of the dose response curve (**Fig. S4 A**), albeit shifted. This suggests that the repressive effect and Hill shape of the curve are primarily dependent on steady-state YAP concentrations. Similarly, and in further support, our optogenetically-controlled experiments include a continuous export condition (chronic light; **Fig.S6A**) that prevents the occurrence of YAP dynamics by continuous YAP export for 12h. For this condition, the steady-state dose response is slightly shifted, but the overall Hill curve shape is again maintained, demonstrating that a concentration-dependent decoding mode is dominating the repressive dose-response function, at least on the time scale of this experiment. To address this point, we edited the main text (pg. 4) as follows:

"While we cannot rule out the occurrence of YAP dynamics under steady-state expression conditions, the evident correlation of nuclear YAP levels with Oct4 and Nanog levels (Fig. 2C, Oct4: $R^2 = 0.96$, Nanog: $R^2 = 0.97$) suggest at least a strong concentration-dependency."

That said, the concerns of this reviewer have been largely addressed.

We thank the reviewer for their input, what has significantly strengthened our manuscript.

Reviewer #3 (Remarks to the Author):

The authors have fully addressed my concerns.

We thank the reviewer for their helpful suggestions and are pleased that we have addressed their concerns in the revision.

Reviewer #4 (Remarks to the Author):

This study quantifies changes in YAP dynamics during the differentiation of naïve mESCs to mesendoderm or ectoderm and perturbs system dynamics using elegant and technically impressive optogenetic control and image-based readouts. The authors report interesting changes in YAP dynamics during differentiation (from sustained to oscillating) and show that different YAP target genes Oct4 and Nanog have different sensitivities to YAP levels (which also depend on the cellular state), that transcription at the Oct4 locus is transient in response to a decrease in nuclear YAP, and that YAP dynamics modulation during differentiation can affect cell fate proportions. This study attempts to explain how different YAP signaling dynamics can direct different target gene and cell fate outcomes, and the results are of broad interest to the fields of optogenetics, signaling dynamics, and developmental biology. However, a few concerns remain regarding the interpretation of results and the lack of experimental support for certain claims of the study.

This review is to specifically assess the authors' responses to Reviewer #2 criticism.

9. The authors added important controls for the quantification of oscillatory YAP dynamics, and particularly the addition of an internal GFP control strengthens their claims.

We are pleased that the reviewer is satisfied with these experiments. We agree that these additional controls significantly strengthen our analyses of native YAP dynamics.

10. a. The added figure S7 provides important controls to support the authors' claim that Oct4 is more strongly induced by an oscillatory YAP rather than a continuous YAP signal. However, could the authors please clarify why the constant 40% YAP export condition in Fig S7C differs greatly from the "chronic" condition in Fig 3C? The data in Fig S7C seems to indicate that comparable Oct4 expression can be achieved both with oscillatory YAP dynamics and 40% "chronic"/continuous light illumination, which would undermine the key conclusion that "oscillatory dynamic YAP inputs more efficiently induce Oct4 and proliferation than do sustained inputs".

We are sorry this point has not been communicated more clearly in the previous revision. When comparing the shape of the Hill curve pre and post illumination, we find that chronic YAP export induces a minor shift of the Hill curve at lower YAP levels (around the IC₅₀ of the Hill curve, **Fig. S6A top left graph**). In contrast, oscillatory YAP dynamics induce Oct4 even at higher YAP concentration, demonstrating higher potency than the chronic condition (**Fig. S6A top right graph**). When we summarize these phenotypes as population averages to compare across several conditions (bar graphs in **Fig.3C, Fig.S6A and Fig.7C**), we bin on two different cell populations to distinguish these differences. We either include all cells that are repressed in the

dark (cells with YAP levels \geq IC5, **Fig. S6B**) or only those that are moderately repressed by at least 50% in the dark (cells with YAP levels \geq IC50, **Fig. 3C**). While oscillatory and chronic inputs are comparable in their potency to induce Oct4 at lower YAP levels (\geq IC5, shown in **Fig. S6B**), oscillatory inputs are significantly better than chronic inputs in the context of higher YAP levels (\geq IC50, **Fig. 3C**). Importantly, for **Fig. S7C**, which the reviewer is referring to and in which we tested additional light doses and light pulses modulations, the quantification includes all cells with lower YAP levels (cells with YAP level \geq IC5 of the dark control). This is because we are comparing among chronic light conditions where the full light dose only shows a phenotype when including lower YAP expressing cells. The chronic vs pulsed (240min ON/60min OFF) condition of **Fig. 7C** is comparable to the previous results using the same cut-off (**Fig. S6B**). Importantly, in **Fig. 7C** we find both specific oscillatory light conditions and chronic low light conditions that are insufficient to induce Oct4 levels even when all cells are included, rejecting the hypothesis that time integrated or intermediate export conditions can account for the observed phenotypes. We edited the Figure legend (**Fig. 3C** and **Fig. S6B**) and the main text to improve clarity and have included the IC5 in **Fig. S6A** in addition to the IC50. The main text (pg.5) now states:

*“IF staining and single-cell quantification of Oct4 protein levels revealed that Oct4 generally shows a minor response to chronic YAP export (**Fig. S6A**, top left graph) but a potent response to oscillatory YAP dynamics (**Fig. S6A**, top right graph). While chronic light only induces Oct4 at lower YAP levels (**Fig. S6B** for cells expressing YAP levels \geq IC5 of dark control), oscillatory light suffices for Oct4 induction even at high levels of YAP (**Fig. 3C** for cells expressing YAP levels \geq IC50 of dark control) at a magnitude that is significantly different from the chronic response.”*

In addition, the legend of Fig.3 C,D now states:

“Quantifications shown only include cells with YAP levels \geq IC50 in the dark condition. Note that the chronic light condition is sufficient to induce Oct4 in cells with lower YAP levels (see Fig. S6 A,B). “

b. Another small comment is that the use of many different terms to describe similar (though sometimes importantly different) conditions is confusing e.g. “light dose modulation” (Fig S7), “sustained” (“Fig 1), “concentrations” (Fig 7), “levels” (Fig 6), “steady-state” (Fig 5), “chronic” (Fig 3). These terms are sometimes used interchangeably (e.g. levels, steady-state, and concentrations). It would help if the authors define these terms clearly and use them consistently in the proper context.

For conditions with optogenetically-induced YAP dynamics, it is important that we properly distinguish the relevant dynamical features such as “light dose modulation”, “chronic light”, “sustained export” etc. We therefore kept the terms but define them when we first introduce them in the main text (pg. 4) as follows:

“We refer to the inducible system as our method for manipulating steady-state concentrations... []... We refer to conditions with continuous light exposure as chronic input with sustained low YAP levels and define conditions with pulsed light exposure as dynamic input with oscillatory YAP dynamics.”

11. The authors have added important controls to support their claims, but a few concerns still remain on the interpretation of results. In general, the manuscript would benefit from clarification and rephrasing to reduce overstatement and speculation. I urge the authors to carefully go

through the claims they are making in the manuscript and correct the wording to match the experimental evidence.

a. Firstly, it is difficult to interpret the results of Figure 2 and Figure 6 without a clearer understanding of the experimental setup. It is stated that the mESCs are undergoing “spontaneous” and “undirected differentiation”. How is this different from the mesoderm and ectoderm differentiation regimes of Fig 1?

The directed mesoderm and ectoderm differentiation protocol makes use of lineage inductive cues (CHIR, mesoderm; retinoic acid, ectoderm). In contrast, the spontaneous differentiation conditions lack these cues in the concentrations required to direct the majority of cells into a specific lineage (spontaneous differentiation media contains FBS). While we use the directed differentiation to compare endogenous YAP dynamics for two different lineages (**Fig. 1**), the spontaneous differentiation conditions were chosen to provide a biochemical environment that is permissive for instructive cues from our inducible and optogenetic systems.

b. What are the cell lineages and cell type proportions under WT unperturbed conditions?

For the cell fate analysis, we used two different spontaneous differentiation media to ensure permissive conditions for specification into all three germ layer fates. Although the undirected differentiation media used throughout the study (FBS-based media, **Figs. 2-4**) permits differentiation into all three germ layers (meso/endo/ectoderm), the ectodermal lineage induction is very low (~1-2%, see IF images in **Fig. S9B**) and potentially restrictive. We therefore probed ectoderm specification in permissive N2B27 media. With these two conditions we detect 26% mesoderm (Tbra positive) and 14% endoderm (FoxA2 positive) specification in FBS-based media, and 28% ectoderm (Sox1 positive) specification in N2B27 differentiation media for unperturbed WT cells. We edited the text (pg.9-10) to explain the rationale for our differentiation conditions as follows:

*“To ensure permissive differentiation conditions for all three germ layer fates, we chose two different spontaneous differentiation media. We used the same FBS based differentiation media as in our previous experiments (**Fig. 2-4**). This condition favors mesendoderm differentiation yielding on average 26% mesoderm (Tbra positive) and 14% endoderm (FoxA2 positive) specification in WT mESCs. In addition, we used N2B27 media which is permissive for the ectoderm fate yielding on average 28% ectoderm (Sox1 positive) for WT mESCs at 5d post differentiation.”*

c. If the cells are undergoing spontaneous differentiation, could it be that the majority of cells develop oscillatory YAP signaling, as in Figure 1? Even though the inducible YAP system expresses YAP at constant dox-controlled gene expression levels, it is conceivable that oscillations (ex. in nuclear YAP amounts via nuc/cyt shuttling) can still emerge during differentiation due to the corresponding changes in cell state. Therefore, is static analysis (as opposed to live imaging) the correct modality for correlating YAP intensity with Oct4/Nanog levels in Figure 2?

Reviewer 1 had a similar concern. We replicate our response here. We cannot rule out that endogenous YAP dynamics contribute to the concentration-dependent effect we observe in Fig. 2. However, the evident correlation (Oct4 $R^2=0.96$; Nanog $R^2 = 0.97$) between nuclear YAP levels and Oct4/Nanog protein levels (**Fig. 2C**) suggest a strong concentration-dependency. In addition, the naïve (2i+LIF) condition, in which we do not detect substantial YAP dynamics (**Fig. 1E**), shows

a similar shape of the dose response curve (**Fig. S4 A**), albeit shifted. This suggests that the repressive effect and Hill shape of the curve are primarily dependent on steady-state YAP concentrations. Similarly, and in further support, our optogenetically controlled experiments include a continuous export condition (chronic light; **Fig.S6A**) that prevents the occurrence of YAP dynamics by continuous YAP export for 12h. For this condition, the steady-state dose response is slightly shifted, but the overall Hill curve shape is again maintained, demonstrating that a concentration-dependent decoding mode is dominating the repressive dose-response function, at least on the time scale of this experiment. To address this point, we edited the main text (pg. 4) as follows:

“While we cannot rule out the occurrence of YAP dynamics under steady-state expression conditions, the evident correlation of nuclear YAP levels with Oct4 and Nanog levels (Fig. 2C, Oct4: $R^2 = 0.96$, Nanog: $R^2 = 0.97$) suggest at least a strong concentration-dependency.”

d. Further, can the different media conditions used in Figure 6 be affecting the cell’s ability to “decode” the inducible/optogenetically controlled YAP dynamics? I recommend the authors to clarify this point in the text and explicitly state the assumptions they are making.

Environmental conditions likely affect the cell’s ability to decode YAP inputs. However, with the exception of the ectodermal fate in **Fig. 6A**, all our inducible and optogenetic YAP experiments shown in **Fig. 2 C,D, Fig. 3C,D, Fig. 4 C,D, Fig.6 A-D** are carried out in the same spontaneous differentiation media conditions (FBS-based media). Cells were furthermore fixed or live-imaged at comparable time windows (1-2d post differentiation start). As explained above in point #11b, for **Fig. 6A** we probed the instructive capacity of YAP levels for ectoderm specification in a different media (N2B27) because the FBS-based spontaneous differentiation media is likely minimally permissive for the ectoderm fate. We also used the N2B27 media to verify that YAP is a pro-proliferative regulator, but this effect is compensated by serum- containing growth factors. As noted above, we edited the text to clarify that differentiation conditions are comparable throughout the study as follows:

“To ensure permissive differentiation conditions for all three germ layer fates, we chose two different spontaneous differentiation media. We used the same FBS based differentiation media as in our previous experiments (Fig. 2-4). This condition favors mesendoderm differentiation yielding on average 26% mesoderm (Tbra positive) and 14% endoderm (FoxA2 positive) specification in WT mESCs. In addition, we used N2B27 media which is permissive for the ectoderm fate yielding on average 28% ectoderm (Sox1 positive) for WT mESCs at 5d post differentiation.”

e. Secondly, the authors repeatedly refer to a cell “decoding” YAP levels, and that “individual YAP target genes differentially decode YAP levels and dynamics”. However, this claim is worded inconsistently throughout the manuscript and seems to lack strong experimental evidence. After Fig 3 the authors state “Together, our results demonstrate that the Oct4 signaling module has two different decoding capacities for steady-state and dynamical inputs”, whereas after Fig 5 the authors state “This suggests that observed differences in the interpretation of steady-state and dynamic YAP inputs are likely established through other regulatory feedback differences, possibly at the level of YAP itself.” Which module, then, is responsible for this differential decoding (confusingly worded as ‘differences in the interpretation’) - is it Oct4 (due to YAP binding to its regulatory regions), YAP itself, or some other cell processes/regulatory feedback? It would greatly improve the clarity of the paper if the authors can clearly explain what they mean by “decoding”

and “differential decoding” and to stay consistent about which process they hypothesize is performing the “decoding”.

We generally refer to “decoding” to describe the cells’ ability to respond to our steady-state concentrations or optogenetic inputs, independent of the underlying decoding mechanism. We term it “differential decoding” when the same YAP input induces different functional outputs in two contexts: (1) When the same YAP levels or oscillatory YAP dynamics differentially affect differentiation and proliferation (**Fig. 6**), and (2) when the same steady-state YAP concentration induces different effects on Oct4 and Nanog (**Fig. 2**).

How nuclear YAP levels or dynamics are sensed by gene regulatory systems is unclear and goes beyond the scope of our study. However, in absence of molecular insight, two different decoding modes can be conceptually distinguished: (1) Mechanisms that act on the core transcription machinery and differentially affect transcriptional bursting features, i.e. the frequency, amplitude or duration of polymerase initiation at the promoter, and (2) those that act through other regulatory modes and affect the magnitude of the same transcriptional bursting feature. We probed YAP’s effects on these two different decoding modules by inferring promoter states from our MS2 live imaging data using a previously-reported hidden Markov model analysis. From our analysis, we find that both steady-state YAP concentrations and dynamics (acute YAP export) primarily affect Oct4 transcription by modulation of the same burst feature (burst frequency). This rejects the hypothesis that the observed potency of oscillatory YAP inputs is established through a switch in transcriptional bursting mode. We note that our previous conclusion “... [] that observed differences in the interpretation of steady-state and dynamic YAP inputs are likely established through other regulatory feedback differences, possibly at the level of YAP itself.” is too speculative and may lead to misinterpretation. Therefore we removed this statement from the text. In addition, we edited the introductory paragraph of the section to better define “decoding” (pg.8). It now reads:

“Our results reveal two different YAP decoding modes by Oct4 that are implemented at the gene regulatory level: steady-state YAP levels control Oct4 in a dose-dependent manner, while acute YAP changes induce an adaptive transient response. How nuclear YAP concentrations or dynamics are sensed (“decoded”) by the Oct4 gene is unclear. Here, we set out to test how YAP interfaces with the underlying gene regulatory network.”

f. On a related topic, could the authors please clarify exactly which experimental evidence allows them to make the central claim that YAP signals are “differentially decoded” by Oct4? From the data presented, it rather supports a conclusion that YAP dynamics are differentially encoded depending on cellular state (proliferation vs. differentiation), and are subsequently consistently decoded via YAP at Oct4 regulatory regions. Indeed, YAP sensitivity is clearly dependent on cell state (Fig S4A vs. 2C), and YAP dynamics change depending on the cell state changes associated with differentiation (Fig 1), which suggests that YAP dynamics are encoded by cell state or environmental cues. It seems the YAP perturbation experiments in Fig 6 are performed during mESC “spontaneous” differentiation conditions, so the question remains: are YAP dynamics instructive to fate specification, or are they permissive and a consequence of (and responding to) fate specification? The broad claims of Fig 7 imply they are instructive cues, and further the authors claim that “cells can titrate YAP doses”. However, can YAP dynamics instead be a response to the broad cellular state changes (signaling, metabolic, cell cycle, mechanics,

epithelial-to-mesenchymal transition) or population context (cell density, cell contacts) that occur during mESC differentiation? Could the differentiation compounds (CHIR, retinoic acid, or FBS) induce differences in YAP dynamics and the observed switch from sustained to oscillatory, without a cell actively “titrating YAP doses”? I would like the authors to comment on these questions in the Discussion section and amend their strong claims in the text accordingly.

The reviewer is correct that the environmental conditions (e.g. differentiating vs naïve mESC) instruct YAP dynamics (**Fig. 1E**). While this is an important finding, our study does not directly address the encoding problem (what upstream signals are encoded in YAP levels or dynamics?). Instead, we focus on the decoding problem of how YAP concentrations and dynamics control downstream decisions (gene activation, cell behavior). To this end, we are keeping the environmental conditions constant (differentiation media, time window during differentiation) and only vary the YAP input (concentrations or dynamics). Under these conditions, we show that the potency of gene activation (Oct4, Nanog) and cell physiology (differentiation, proliferation) is dependent on the YAP input, demonstrating that YAP levels and dynamics are instructive cues. This supports the schematic summary in **Fig. 7**. In response to point #11d, we modified the text to clarify that the differentiation media/conditions are kept constant. We appreciate the request for a more detailed discussion of the encoding problem and have added the following new paragraph in the discussion to acknowledge this exciting question (pg. 12):

“While our study focuses on the decoding logic of YAP levels and dynamics, our finding that differentiating mESC exhibit sporadic YAP pulses poses exciting new questions about information encoding through YAP: What information is encoded in YAP levels and dynamics? Cellular differentiation is accompanied by substantial changes in cellular metabolism and morphology for example from cell-cell or cell-ECM adhesion, alterations of cell density, the establishment of cell polarity, and changes in the mechanical properties of the environment (e.g. ECM deposition, hydraulic pressure of the blastocyst lumen). Given that YAP is mechanoresponsive, it is tempting to speculate that different cellular mechanical inputs may be encoded in YAP dynamics or concentrations. Coupling reporters for YAP dynamics with different mechanical inputs could help address this question.”

REVIEWERS' COMMENTS

Reviewer #4 (Remarks to the Author):

The authors have clarified the text and addressed my concerns. One small comment is, could the authors please clarify in the figure legend what is the "YAP control" condition shown in Fig. S6A?

Point-by-point response to the Reviewer

We thank the Reviewer for the additional feedback. Below we provide a response to the remaining question.

REVIEWERS' COMMENTS

Reviewer #4 (Remarks to the Author):

The authors have clarified the text and addressed my concerns. One small comment is, could the authors please clarify in the figure legend what is the “YAP control” condition shown in Fig. S6A?

The YAP control cells express the same SNAP-YAP construct as the LEXY-SNAP-YAP cells but lack the light-sensitive LEXY-tag. We use it to control for light-induced artifacts in our experiments. We added a sentence in the Figure legend to clarify this point (highlighted in red in the legend below).

Figure S6 Quantification of Oct4 induction upon light gated control of YAP export

A) Sigmoidal curve representation of the light-gated Oct4 induction shown in Fig. 3C. Shown are Oct4 protein levels as a function of nuclear LEXY-YAP/YAP levels upon chronic (top left) and pulsed (240min ON/60min OFF; top, right) illumination in LEXY-YAP (top) and YAP control cells (bottom). **The YAP control cells express the same SNAP-YAP construct as the LEXY-YAP cells but lack the light-sensitive LEXY-tag.** All light conditions (blue) are shown in reference

to a dark control (grey). Shown are mean from N=7 independent experiments and the sigmoidal curve fits with 95% CI. Dashed lines indicate the IC5 and IC50 of the dark control condition. Red arrows indicate Oct4 induction upon pulsed illumination (broad range of YAP levels) or chronic light (small range of YAP levels). **B**) Quantification of Oct4 protein levels upon chronic or pulsed (240 min ON/ 60min OFF) light as shown in **Fig. 3C**. The quantification includes all cells expressing YAP levels \geq IC5 (see grey dashed line in panel A). Note that the quantification of Oct4 levels in higher YAP expressing cells (\geq IC50) shows higher potency of oscillatory than chronic YAP inputs (see **Fig. 3C**). Shown are mean \pm SEM, N=7 independent experiments. P values from unpaired Student's t test. **C**) Relation of our YAP-dependent Oct4 phenotypes to the range of endogenously-observed Oct4 levels. Following pluripotency exit, mESCs decrease endogenous Oct4 levels by \sim 73% over a time course of 4 days post spontaneous differentiation. Our steady-state measurements of Oct4 repression (**Fig. 2C**) compares to \sim 83% (low YAP levels) and 4% (high YAP levels) of the Oct4 protein found in naive cells (indicated by horizontal dashed line, red shading represents the SEM of that measurement, N=3). The magnitude of our optogenetic Oct4 induction through oscillatory YAP dynamics (**Fig. 3C**, pulsed light, 240min ON/ 60min OFF, Δ Oct4 = 0.25 \pm 0.1) is comparable to the amount of Oct4 protein lost within 1d during spontaneous differentiation. Oct4 levels were quantified from IF stainings. Shown are mean \pm SEM, N= 3 independent experiments.